# A nanobody specific to prefusion glycoprotein B neutralizes HSV-1 and HSV-2

Benjamin Vollmer[1,2,3], Henriette Ebel[1,2,3], Renate Rees[4], Julia Nentwig[1,2,3], Thomas Mulvaney[1,3,5,6], Jürgen Schünemann[4], Jens Krull[4], Maya Topf[1,5,6], Dirk Görlich[4✉] & Kay Grünewald[1,2,3✉]

The nine human herpesviruses, including herpes simplex virus 1 and 2, human cytomegalovirus and Epstein–Barr virus, present a significant burden to global public health[1]. Their envelopes contain at least ten different glycoproteins, which are necessary for host cell tropism, attachment and entry[2]. The best conserved among them, glycoprotein B (gB), is essential as it performs membrane fusion by undergoing extensive rearrangements from a prefusion to postfusion conformation. At present, there are no antiviral drugs targeting gB or neutralizing antibodies directed against its prefusion form, because of the difficulty in structurally determining and using this metastable conformation. Here we show the isolation of prefusion-specific nanobodies, one of which exhibits strong neutralizing and cross-species activity. By mutational stabilization we solved the herpes simplex virus 1 gB full-length prefusion structure, which allowed the bound epitope to be determined. Our analyses show the membrane-embedded regions of gB and previously unresolved structural features[3,4], including a new fusion loop arrangement, providing insights into the initial conformational changes required for membrane fusion. Binding an epitope spanning three domains, proximal only in the prefusion state, the nanobody keeps wild-type HSV-2 gB in this conformation and enabled its native prefusion structure to be determined. This also indicates the mode of neutralization and an attractive avenue for antiviral interventions.

Recent advances in the development of subunit vaccines have shown the potential of prefusion-stabilized viral membrane fusion glycoproteins to elicit neutralizing activity[5–15]. The rationale is that the membrane fusion proteins on the infectious virus necessary to merge the viral with the host cell membrane are in prefusion conformation, exposing epitopes unique to this structure that can be targeted by neutralizing antibodies to prevent infection before the virus enters the cell. Likewise, glycoproteins on newly produced viral particles are recognized by these antibodies. The extensive domain rearrangements involved in the fusion process indicate a substantial change in the exposed surface giving rise to conformation-specific epitopes. Targeting these epitopes allows intervention by impairing membrane fusion, thus achieving neutralization. For herpesviruses, first steps have been made to stabilize the membrane fusion protein glycoprotein B (gB)—the most conserved glycoprotein in all herpesviruses[2]—to obtain its prefusion structure[16,17]. So far there is only one high-resolution prefusion structure of gB, namely from the beta-herpesvirus human cytomegalovirus (HCMV), that was captured in this state using a specific inhibitor and chemical crosslinking and purified using an antigen-binding fragment (Fab) of a neutralizing antibody[4]. The structure matches previous low-resolution data[18], but shows differences from herpes simplex

virus 1 (HSV-1) gB[16]. This could indicate structural variations between alpha- and beta-herpesviruses, or that these proteins were captured in different states[19], pointing to dynamic and energetic differences. Furthermore, to design effective stabilizing mutations that make these metastable proteins amenable for vaccine and antiviral development, more high-resolution structures of prefusion gB are required.

A range of monoclonal antibodies directed against HSV-1 gB is available, targeting different epitopes, but often lacking virus-neutralizing activity[20–22]. Because of the metastability of gB, antibody generation as well as epitope mapping traditionally used peptides or constructs presenting the ectodomain in postfusion conformation[21,23]. Consequently, mechanistic insights into the modes of action of these antibodies are lacking. Particularly missing for clinical applications are antibodies specifically targeting the prefusion conformation of gB[22].

Recently, the great potential of neutralizing nanobodies as therapeutic agents directed against the SARS-CoV-2 spike protein was demonstrated[24,25]. Nanobodies are single-domain antibodies derived from camelid heavy chain-only antibodies that feature a single-domain binding site and can be cloned and expressed individually. In comparison with conventional monoclonal antibodies that have to be produced mostly in eukaryotic cells, nanobodies can be produced efficiently

[1]Centre for Structural Systems Biology (CSSB), Hamburg, Germany. [2]Department of Chemistry, University of Hamburg, Hamburg, Germany. [3]Department of Structural Cell Biology of Viruses, Leibniz Institute of Virology (LIV), Hamburg, Germany. [4]Department of Cellular Logistics, Max Planck Institute for Multidisciplinary Sciences, Göttingen, Germany. [5]University Medical Center Hamburg-Eppendorf (UKE), Hamburg, Germany. [6]Department of Integrative Virology, Leibniz Institute of Virology (LIV), Hamburg, Germany. ✉e-mail: goerlich@mpinat.mpg.de; kay.gruenewald@cssb-hamburg.de

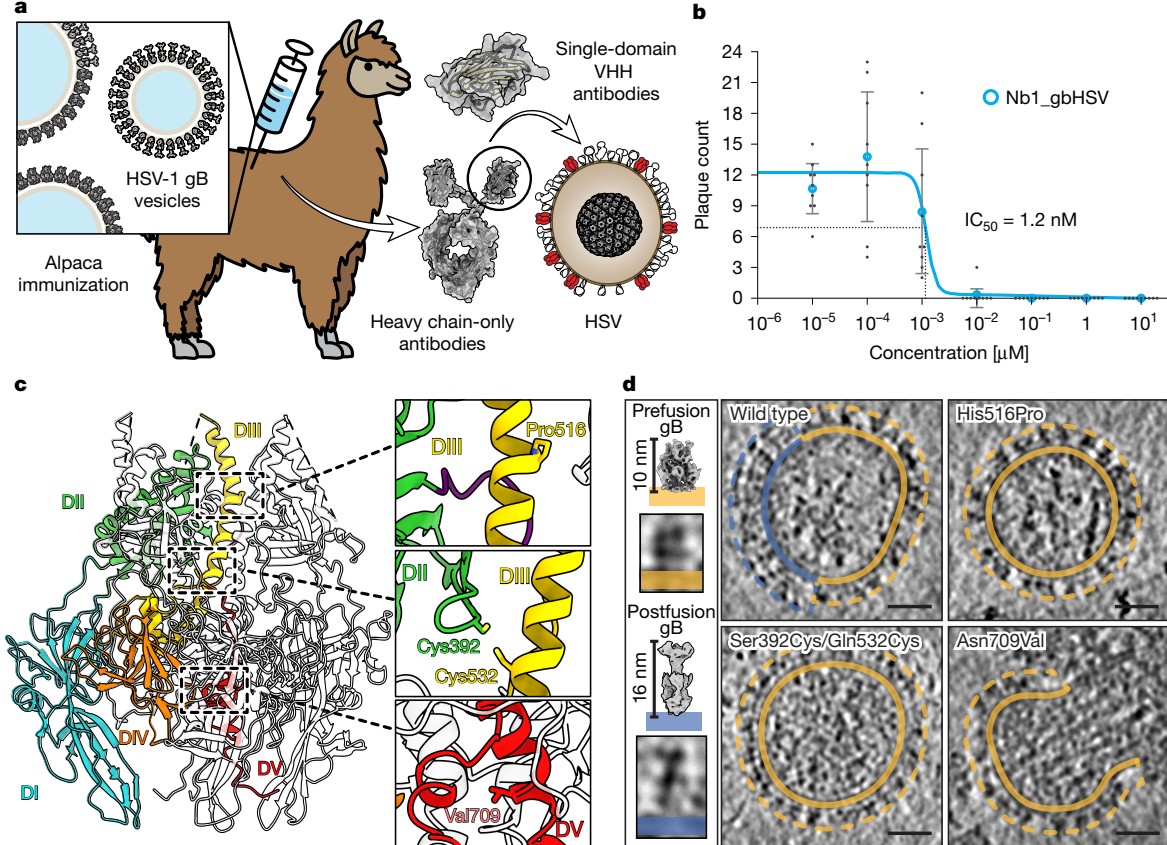

**Fig. 1 | Nanobody generation and target stabilization. a**, Graphical illustration of anti-gB nanobody generation and testing. Alpacas were immunized using gB vesicles, and selected nanobodies were expressed and tested for HSV-1 neutralization. The camelid heavy-chain-only antibody illustration was generated using Protein Data Bank (PDB) 9Q9N chain d for the VHH regions, grafted onto the immunoglobulin-γ framework of PDB 1IGT (ref. 46). For the virus illustration PDB 6CGR was used as capsid[47]. **b**, $IC_{50}$ determination in plaque reduction assays using a 10 pM to 10 μM concentration range. Each blue point represents the average of $n = 3$ independent experiments with $n = 3$ technical replicates each. Individual data points are marked as grey dots. Error bars, s.e.m. **c**, Side view of HSV-1 gB pseudoatomic model (PDB 6Z9M)[16] used to design mutations with marked domains coloured in one promoter. Individual mutations are shown in insets with residues as sticks with atoms in blue for nitrogen, yellow for sulfur and red for oxygen. **d**, Representative vesicles shown from cryo-ET slices of one of at least $n = 2$ independent, successful production and purification of gB vesicles with the described mutations. Membrane regions (lines) and maximum protein extension (dashed) are coloured according to the conformation. Exemplary densities for the two conformations (orange, prefusion gB; blue, postfusion gB) are shown on the left. Scale bars, 25 nm.

and cost-effectively in *Escherichia coli* or yeast cells. Due to their small size and rigidity, non-conventional delivery routes, including topical or aerosolized delivery, are possible[25].

Here we report a prefusion-specific nanobody against gB that is neutralizing with cross-species activity. We determined full-length high-resolution cryo electron microscopy (cryo-EM) structures of gB in its prefusion and postfusion conformation and describe the nanobody epitope and mode of neutralization. The topology of the conformational epitope indicates that binding of the nanobody prevents the conformational change to the postfusion form necessary for membrane fusion, which in turn would explain the neutralizing activity. Moreover, we demonstrate several prefusion-specific, but non-neutralizing, nanobodies that target a different epitope at the apex of prefusion gB, indicating that prefusion specificity does not necessarily imply neutralizing activity.

### Nanobody generation against gB

To generate gB-specific and preferably prefusion-specific nanobodies, we immunized alpacas (Fig. 1a). However, purified gB, when removed from the membrane or when omitting the transmembrane regions in ectodomain constructs, results in gB irreversibly adopting the postfusion conformation and thus unable to elicit prefusion-specific nanobodies. To preserve the prefusion conformation, we purified gB

in extracellular vesicles[26,27]. On the vesicle surface the protein is found in both prefusion and postfusion conformation. After collecting supernatants of gB transfected cells, vesicles were concentrated and purified for use in alpaca immunizations (Fig. 1a). After RNA isolation from lymphocytes, isolated variable domain of heavy-chain-only antibody (VHH)-coding regions were amplified and cloned into phagemids to generate immune libraries followed by phage display[24]. gB vesicles were used as bait to select clones, which were sequenced and cloned into expression vectors[28]. Nanobodies were tested for their potential to neutralize HSV-1 in plaque reduction assays. Two out of 17 tested nanobodies showed inhibition of plaque formation (Extended Data Fig. 1a). Of these two, only one (Nb1_gbHSV) showed consistent inhibition in low concentrations; its half maximal inhibitory concentration ($IC_{50}$) was determined subsequently to be 1.2 nM (Fig. 1b).

### Design of prefusion stabilizing variants

To determine the structural epitope of Nb1_gbHSV, we introduced stabilizing point mutations in gB to restrict the protein from changing to the postfusion conformation, even after membrane lipid removal. On the basis of insights gained from our previous model of HSV-1 gB, which was determined using a stabilizing helix breaker mutation (H516P) in the central helix of domain III (DIII)[16], we devised mutations that use

different means of stabilization[29] (Fig. 1c). Electron cryo tomography (cryo-ET) of secreted gB vesicles permits direct visual feedback of the conformational state of gB[16,27] (Fig. 1d). First, we selected residues sufficiently close in our model for potential disulfide bond formation between domains that are located in close proximity in prefusion, but separated in postfusion conformation. We devised a set of four double mutants, one of which showed cell surface expression, detected by fluorescently labelled Nb1_gbHSV (Extended Data Fig. 1b,c). Cryo-ET analysis of the purified vesicles containing this mutant, albeit present at a much lower abundance compared with wild-type gB (Extended Data Fig. 1d), confirmed the mutated protein on the surface to be only in prefusion conformation (Fig. 1d). During the prefusion to postfusion transition, the C-terminal part of domain V (DV), connected to the lipid-embedded membrane proximal region (MPR), undergoes a conformational change from a helical bundle at the centre of the trimer (Fig. 1c) to a long, partially helical extension. The same region was implicated to have a role in fusion regulation in the gB homologue of Suid alpha herpesvirus[30]. The residue sequence of this helix follows in part the heptad sequence of coiled-coils (abcdefg− with a and d being hydrophobic residues) with one conserved asparagine at position 709 breaking this sequence. To re-establish the heptad pattern to strengthen the hydrophobic interaction between the helices, N709 was mutated to valine (Fig. 1c). The mutation had no influence on gB expression and vesicle formation (Extended Data Fig. 1d), and stabilized the protein in the prefusion conformation on vesicles (Fig. 1d and Extended Data Fig. 1e). To ensure the stability of the prefusion conformation of gB even without a membrane environment, all four stabilizing mutations were combined in one construct (Extended Data Fig. 1f,g).

## Structure of stabilized HSV-1 gB

The stabilized construct was expressed in stably transduced HEK293T cells[31]. After solubilization with detergent, affinity purification and reconstitution in peptidiscs[32], the protein was purified by size exclusion chromatography (Supplementary Fig. 1). Concentrated peak fractions were used for single-particle cryo-EM analysis. Samples were prepared with and without preincubation of gB with Nb1_gbHSV. Two-dimensional classification of the picked particles showed gB in a conformation different from the postfusion form (Extended Data Fig. 2). During processing, no classes showing gB in postfusion conformation were observed, demonstrating that the mutations fully stabilize the protein. We resolved the prefusion structure of stabilized gB at an overall resolution of 2.74 Å (Extended Data Fig. 2c) whereas the nanobody-bound structure reached 3 Å (Extended Data Fig. 3d and Supplementary Fig. 2). Notably, in both structures, the resolution of membrane-embedded domains is substantially lower than the ectodomain, which indicates an intrinsic flexibility of these regions even in the stabilized protein (Extended Data Figs. 2d and 3d). When comparing the prefusion and postfusion conformation, several structural changes in the domains are evident, alongside the rearrangement of the individual domains as a whole (Fig. 2). The previously unresolved N-terminal region encompassing residues 90–106 forms a helix that is held in the groove between domain I (DI) and domain II (DII) of the preceding protomer that is formed by repositioning of a short loop region (amino acids 407–410) in the connection between β20 and αB[33] in DII (Fig. 2c). This arrangement creates a hydrophobic surface on DII formed by residues Y402, L404, V407 and L443 against which the helix is held by orienting a hydrophobic side formed by residues L97, L101 and I104 towards DII. In addition, R98 and K105 form two salt bridges with E152 located in the region connecting DI and DII.

The central helix (αC) in DIII contains an N-terminal loop region (501–510), reaching over DII at the apex of gB (Fig. 2b,d). This loop transitions during the fusion process to form the fully extended helix (501–545) that creates the central three-helix bundle in the postfusion conformation[33]

(Fig. 2d). Preceding αC, connected by an unresolved linker, is αX of DII, which is in a position perpendicular to its arrangement in the postfusion structure[34] (Fig. 2d). The first crystal structure of gB from HSV-1 (ref. 33) in the postfusion conformation, as well as structures from varicella zoster virus[35] (VZV), HCMV[36] and Epstein–Barr virus[37] (EBV) did not resolve αX, indicating a high flexibility in this state. The upright position of αX seems necessary to accommodate the loop region of DIII, as they would otherwise clash if αX was lying flat on DII as in the postfusion conformation. The αC loop region harbours several hydrophobic residues that are well conserved, with Y510 being fully conserved in all herpesviruses. These hydrophobic residues form a patch that might be necessary to hold the αX helix in place. The αX helix was recently resolved in a prefusion-stabilized ectodomain construct of HCMV gB[17], showing an extra orientation, perpendicular to the position found in the postfusion conformation of gB[34]. Similar to the postfusion arrangement, this position would most probably clash with the N-terminal loop region of the DIII helix observed in our structure (Fig. 2b,d), but can be explained by the fact that, in HCMV gB, the central helices are already fully extended in the prefusion conformation[4] (Extended Data Fig. 4).

The tips of DI form the fusion loops that are inserted into the target membrane during infection. Unlike class I membrane fusion proteins, which contain an N-terminal hydrophobic fusion peptide, the first crystal structure of postfusion gB showed a set of two internal loops connecting three beta strands and forming the tip of DI (Fig. 2e). This arrangement is reminiscent of class II fusion proteins where internal loops connecting beta strands form the fusion loops. However, alpha- and flaviviruses contain only one loop, whereas the hantavirus Gc protein contains a tripartite version[38]. In the postfusion trimer of gB, the residues of the fusion loops (amino acids 174–179 and 259–262) form a hydrophobic ridge that is surrounded by charged residues[39]. Using three-dimensional classification and local refinement, we resolved the fusion-loop-containing region and showed large differences between the pre- and postfusion conformation (Fig. 2e). In the postfusion conformation, the extended fusion loops are located in a short loop connecting the beta hairpin formed by β4 and β5 (fusion loop I) and in the loop directly preceding β11 (fusion loop II)[33,34] (Fig. 2e). The side chains of the fusion loop residues point towards the membrane-embedded MPR of the adjacent protomer, although a direct interaction with the residues of the MPR was not seen for HSV-1 gB as the first of the two MPR helices was not resolved previously[3]. In the prefusion structure of HCMV gB, the fusion loops adopt the same extended arrangement (Extended Data Fig. 4), forming interactions with the MPR. By contrast, in prefusion HSV-1 gB, the three beta strands (β4, β5 and β11) are significantly (four to six residues) shorter. Fusion loop I therefore changes its length from four to 15 residues and 'rolls up', forming a short (2 turn $3_{10}$) helix while keeping contact with the MPR by positioning W174, F175, Y179, F182 and M183 towards the hydrophobic cleft of the MPR (Figs. 2e and 3a,b). In addition, a hydrogen bond is formed between Q172 and E740 of the MPR (Fig. 3a). The region containing fusion loop II also diverges from the postfusion conformation starting with residue D251 and forming a $3_{10}$ helix (S257–Y265) that is stacked on top of the helix formed by fusion loop I (Fig. 2e). F262 of fusion loop II is thereby nested in a hydrophobic pocket, created by residues I185, L252 and residues L686 and Y689 (DV) of the adjacent protomer (Fig. 2e). Alignment of our described fusion loop 'rolled up' form with the extended fusion loops of the postfusion conformation shows clashes of the latter with the MPR, in particular amino acids A261 and F262 of fusion loop II of the extended form. During the fusion process the dissociation of the fusion loops from the MPRs might trigger the conformational change into the extended form, before being inserted into the target membrane (Fig. 2e).

## Structure of membrane-embedded region

Anchored in the upper region of the lipid bilayer, two amphipathic helices form the MPR, which extends below DI before turning back

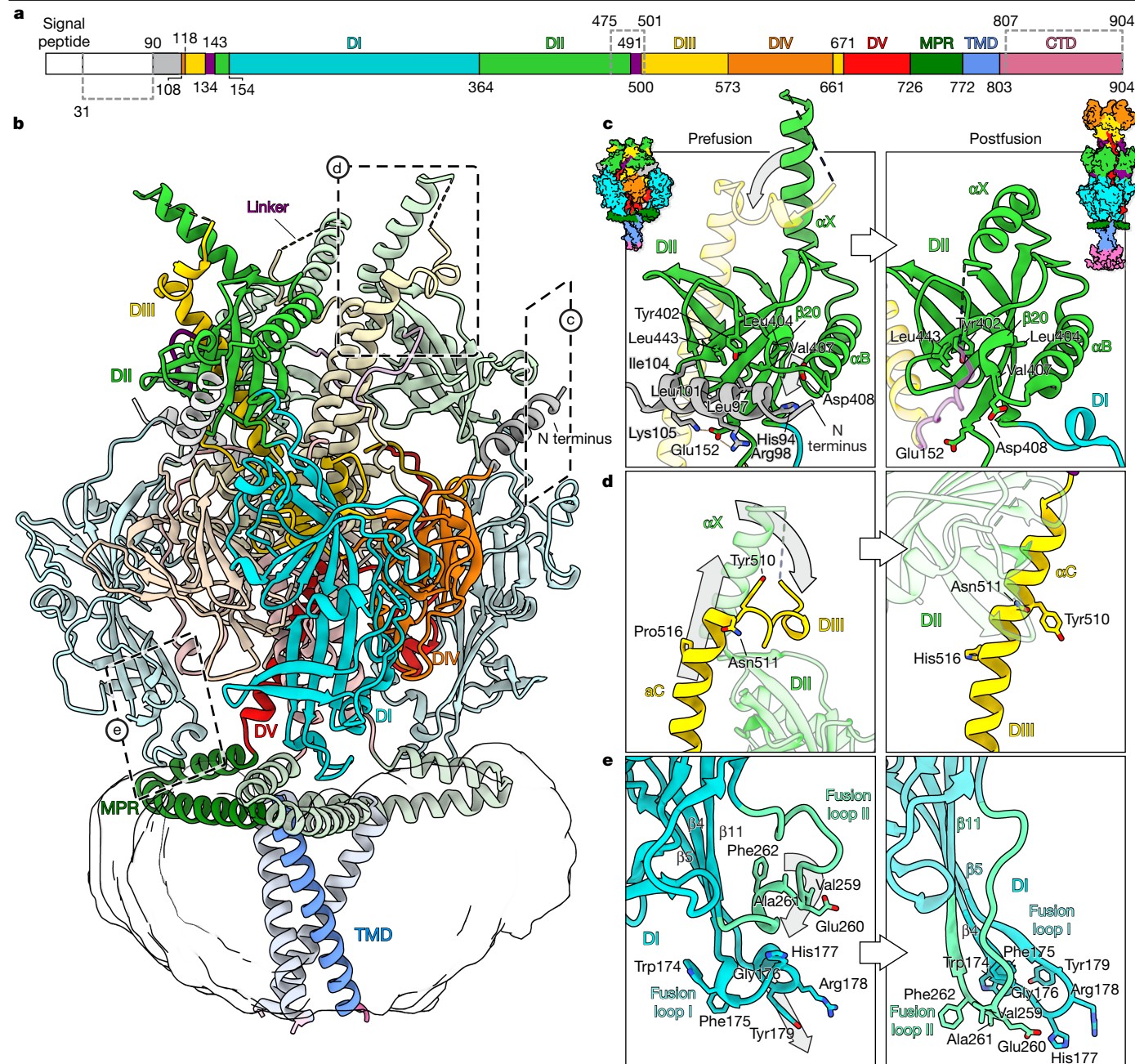

**Fig. 2 | Full-length prefusion structure of gB. a**, Domain architecture of gB with numbers indicating amino acid positions of the domain boundaries. N-terminal signal peptides and the unstructured N-terminal domain of gB are shown in white, flexible linker regions in purple. Regions not resolved in the structure are marked with dashed lines. **b**, gB structure shown in ribbon style with domains marked by Roman numerals and coloured as in ref. 35. TMD and MPR are coloured as in ref. 3. Region and viewing angle of the other panels are indicated by dashed boxes. **c**–**e**, Detailed views of the structural features in domains with the prefusion conformation shown in left boxes and postfusion conformation (PDB 5V2S)[3] shown in right boxes, as indicated by the structural pictograms at the top. Moving regions are indicated by grey arrows. Selected residues are marked in three-letter code and shown in sticks with atoms shown in blue for nitrogen and red for oxygen.

towards the centre of the trimer where the connection to the transmembrane domain (TMD) is located (Figs. 2b and 3a). Comparing the prefusion and postfusion arrangement of the MPR and the TMD, the overall structure stays the same. When aligned to the TMD, the in-plane angle between MPR and TMD changes by around 19.5° (Extended Data Fig. 5). The wedge-like shape formed by the MPR shows interactions with fusion loop I of DI of the next protomer. The first amphipathic helix of the MPR forms the connection to DV and is arranged with the hydrophobic side (containing residues M731, F732, L735, F738, F739) facing the hydrocarbon core of the membrane.

Residues F732, L735 and F739 point to M754, V757 and V761 of the second MPR helix to form a hydrophobic cleft (Fig. 3a,b). The underside of fusion loop I forms a hydrophobic footprint that matches the hydrophobic cleft of the MPR, possibly holding DI in place (Fig. 3c). The second helix of the MPR harbours a serine spine (S762, S765, S769) that marks the top side, facing the hydrophilic headgroups of the outer viral membrane leaflet[3] (Fig. 3a,b). The cytosolic/intraviral domain was not resolved in our structure, probably due to the missing membrane surface where the C-terminal amphipathic helix h3 would be embedded[3].

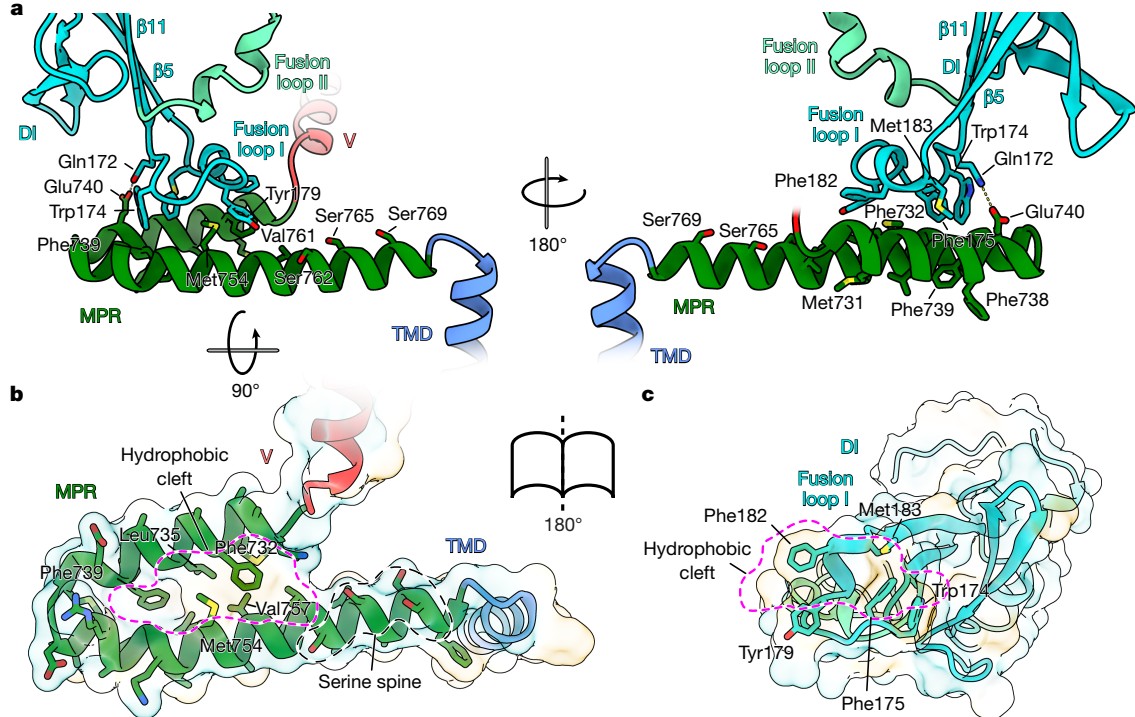

**Fig. 3 | Membrane-embedded regions. a**, Side views of the MPR forming hydrophobic interactions and hydrogen bonds with fusion loop I of DI. **b**, Top view of the MPR helices with residues forming the hydrophobic cleft marked in pink and the serine spine in black dashed lines. Hydrophobicity is shown in half-transparent surface rendering. For clarity, DI is not shown. **c**, Bottom view of fusion loop I with residues forming the hydrophobic footprint and hydrophobic surface. Structures are shown in ribbon rendering and selected residues are marked in three-letter code and shown in sticks with atoms shown in blue for nitrogen, red for oxygen and yellow for sulfur. The MPR is not shown to allow the hydrophobic footprint of fusion loop I to be seen clearly. Hydrophobicity is shown ranging from dark cyan (most hydrophilic) to white to gold (most lipophilic).

## Nb1_gbHSV binds an interdomain epitope

During processing of the nanobody-bound gB dataset, two-dimensional classification showed gB with additional densities, most obvious in top views (Extended Data Fig. 3b). Compared with the overall resolution of the nanobody-bound structure (3 Å) the local resolution at the nanobody binding site was in the range of 2.3–2.8 Å (Extended Data Fig. 3d). The nanobody interacts with three different domains simultaneously (Fig. 4a,b). The interaction features a total of 23 hydrogen bonds and eight salt bridges distributed over separate interaction surfaces on domain IV (DIV) and DI of the adjacent protomer, resulting in 1,260 Å² of buried surface area on gB (Fig. 4c–e and Extended Data Table 1). Furthermore, R102 of Nb1_gbHSV reaches into the gap between DIV and DI, forming hydrogen bonds with DI and DIII of the same protomer. DI and DIV are located close to each other in the prefusion state (Fig. 2b), but are at separate ends in the postfusion conformation, which indicates that the bound epitope, and therefore Nb1_gbHSV, is prefusion-specific. When mapped onto the postfusion structure while remaining bound to its main interaction surface on DI, extensive steric clashes are detected between Nb1_gbHSV, DV and DI of the adjacent protomer (Supplementary Fig. 3). Although part of the epitope on DIV remains accessible also in postfusion conformation, a stable interaction seems unlikely. To determine the affinity of Nb1_gbHSV to prefusion and postfusion gB we used microscale thermophoresis. In this assay, Nb1_gbHSV bound the prefusion form with a dissociation constant ($K_D$) of approximately 14 pM, whereas no affinity was detectable for postfusion gB (Supplementary Fig. 4a). Although sequence alignment of gB of different human infecting herpesviruses (HSV-1, HSV-2, VZV, HCMV, EBV, KSHV, HHV-6 and HHV-7) showed only low conservation of the bound epitope across subfamilies (Supplementary Fig. 5), the epitope is conserved in HSV-2 gB. To test the binding specificity, a construct of Nb1_gbHSV carrying an extra cysteine residue was expressed, purified and labelled fluorescently using Alexa 647 (ref. 40). Indeed, BHK-21 cells transfected with C-terminally superfolder enhanced green fluorescent protein (sfEGFP) tagged versions of HSV-1 as well as HSV-2 gB were bound by the labelled nanobody, as evident by the co-localization of the two fluorescent signals at the plasma and intracellular membranes (Extended Data Fig. 6a,b). The latter is most probably caused by reinternalization of gB by endocytosis after reaching the plasma membrane. Nb1_gbHSV is therefore able to bind gB from both herpes simplex species, demonstrating cross-species reactivity, most probably due to the high sequence similarity. By contrast, although structurally conserved, gB of neither VZV, HCMV nor EBV showed interaction with the nanobody as no fluorescent signal co-localization above the negative control level (sfEGFP-tagged vesicular stomatitis virus glycoprotein G) was detected (Extended Data Fig. 6b). The ability of the nanobody-bound gB protein to be reinternalized into the cell indicates that the interactions with the cellular machinery necessary for endocytosis are not affected.

## Prefusion specificity and neutralization

Using grating-coupled interferometry (Supplementary Fig. 4b–g), we confirmed the picomolar binding affinities of Nb1_gbHSV ($K_D$ = 94 pM) to prefusion, whereas no binding was detected against the postfusion form. To validate the experimental setup, we used a minimal binding fragment of the (co-)receptor PILRα[41] (Supplementary Fig. 6) showing micromolar binding affinity for both prefusion and postfusion gB, as well as Fab fragments of the strongly neutralizing monoclonal antibody SS55 (ref. 21), which also interacted with both conformations, but with nanomolar affinities. To understand why other nanobodies lack a neutralizing activity, we determined the conformation specificity of three nanobodies (Nb2_gbHSV, Nb3_gbHSV and Nb4_gbHSV), representing different neutralization

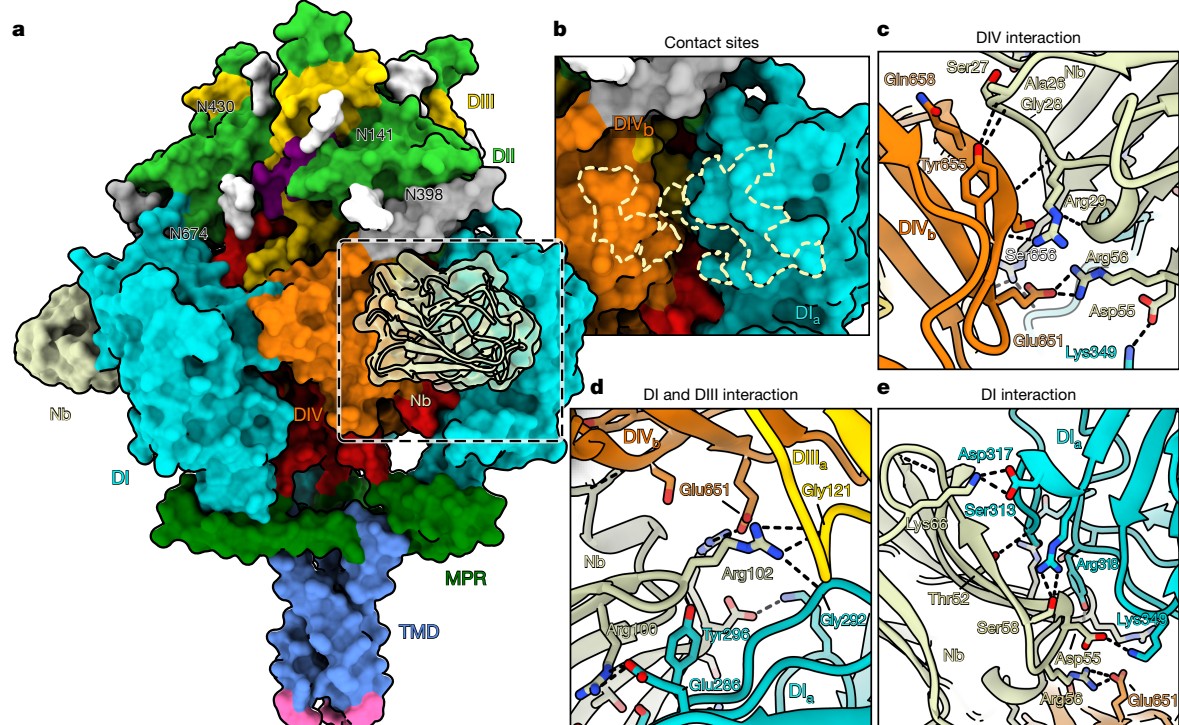

**Fig. 4 | Nb1_gbHSV binds gB. a**, Surface rendering of gB bound by Nb1_gbHSV. Facing the front, Nb1_gbHSV is shown with semi-transparent surface showing the ribbon representation. The region for the close up view of the nanobody footprint is indicated by a dashed box. Glycosylation sites are shown in white with the corresponding residue marked. **b**, Focus on the nanobody contact on gB. Viewing angle is slightly turned and the nanobody structure is omitted for better visibility. The region contacted on gB by the nanobody is outlined by dashed lines. Protomers are indicated by lower case letters a–c on domain numbers. **c**–**e**, Ribbon rendering of the nanobody-bound gB structure, zoomed in on the interaction between Nb1_gbHSV and DIV (**c**), DI and DIII (**d**), and DI (**e**). Hydrogen bonds are marked by dashed black lines and residue names are given in three-letter code with the corresponding position.

activities, using grating-coupled interferometry. All three showed nanomolar binding affinities for prefusion gB ($K_D$ = 46 nM, 0.9 nM and 17 nM, respectively), but no measurable affinity to postfusion gB. Despite their strong binding preference for prefusion gB, these nanobodies are not neutralizing. To understand the structural difference between the neutralizing and non-neutralizing nanobodies, we collected cryo-EM datasets of a mixture of prefusion-stabilized and wild-type (postfusion) gB incubated with a 50-fold excess of the aforementioned nanobodies to allow even low affinity interactions. Difference maps to the apo forms of gB pinpointed the individual binding regions on prefusion and postfusion gB (Extended Data Fig. 6c–g). Notably, for all non-neutralizing nanobodies, clear densities were localized to the αX helix of DII on top of the prefusion structure. By contrast, the flexibility of this helix in the postfusion conformation might contribute to the observed, much lower, binding affinity and occupancy differences (Extended Data Fig. 6e–g). In terms of crowding, the surface of the densely gB-studded vesicles used for immunization (Fig. 1d) roughly resembles the abundance of the collective glycoprotein species on virions. Hence, the apex region would potentially be the most accessible, explaining why most of our nanobodies are directed against this region. There are six N-linked glycosylations in HSV-1 gB[42], of which four could be modelled in our electron microscopy density map (Extended Data Fig. 7). Their distribution protects most of the apex region, while leaving the αX helix exposed (Fig. 4a). For the postfusion conformation, the αX helix was shown to be closely located to N141 and N398 (ref. 3). Hence, fully branched glycosylations on these residues could partially hinder nanobody interactions. Taken together, the observed difference between neutralizing and non-neutralizing nanobodies seems to be a combination of high affinity and specificity to the prefusion form of gB, and the targeted epitope.

## Neutralizing activity of Nb1_gbHSV

The bound epitope spread over different domains that are close in prefusion conformation indicates that the neutralizing activity is achieved by preventing gB from undergoing the necessary conformational changes for the fusion process. To support this hypothesis, and to show the cross-species activity, we co-expressed wild-type HSV-2 gB with the nanobody in stable transduced HEK293T cells[31]. gB is a type I transmembrane protein, presenting the ectodomain at the N-terminal end to the extracellular space and hence during expression into the endoplasmic reticulum lumen. For Nb1_gbHSV to reach its target, an N-terminal signal peptide was added to ensure translocation into the endoplasmic reticulum lumen during translation. The expressed protein was purified as described for the stabilized gB protein. After reconstitution in peptidiscs and purification by size exclusion chromatography, the concentrated peak fraction was used for cryo-EM analysis (Supplementary Fig. 7).

After particle picking and two-dimensional classification, several classes were produced with gB in the prefusion state, resembling our stabilized HSV-1 gB structure, in addition to classes showing the easily identifiable postfusion conformation in side and top views. Close inspection of the prefusion top view class shows the bound nanobody (Extended Data Fig. 8b; blue arrows) as seen for HSV-1 gB (Extended Data Fig. 3b). We then determined the structures of the prefusion and postfusion conformations of HSV-2 gB (Fig. 5b) to 2.85 Å and 2.26 Å resolution, respectively (Extended Data Figs. 8 and 9). The postfusion conformation shows the same structural features as seen for HSV-1 gB, with a root-mean-square deviation of 0.599 Å between the final model and the published structure[33] (Supplementary Fig. 2b). The membrane-embedded regions including the MPR, the TMD and the intraviral C-terminal domain are not resolved in either of the two

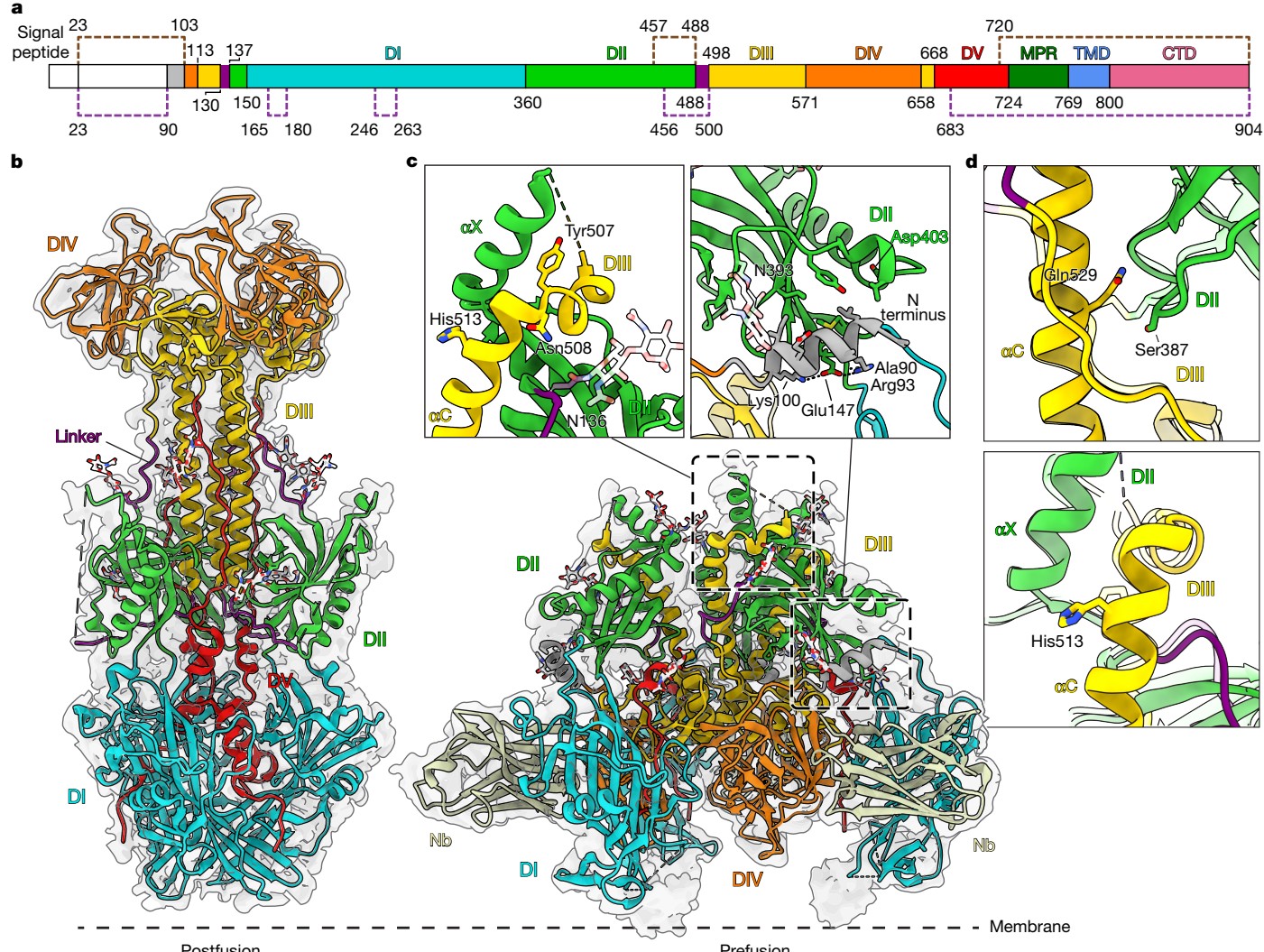

**Fig. 5 | HSV-2 gB bound by Nb1_gbHSV. a**, Linear diagram of the domain architecture of HSV-2 gB as in Fig. 2. Regions not resolved in the structure are marked with dashed lines in brown for postfusion on top and in violet for prefusion gB below. **b**, Postfusion structure of HSV-2 gB in ribbon style rendering, overlaid with the semi-transparent electron microscopy density (contour level 0.0825). Domains are marked by Roman numerals. Location of the membrane is indicated by a dashed line. Modelled glycosylations are shown in white. Atoms in residues shown in stick style with blue for nitrogen, red for oxygen and yellow for sulfur. **c**, Prefusion structure of HSV-2 gB shown as in **b** with insets showing detailed views of structural features. **d**, Regions that were mutationally stabilized are shown. Transparent overlays show the stabilized HSV-1 gB structure in comparison with wild-type HSV-2 gB.

HSV-2 gB structures. The prefusion structure contains most of the ectodomain, with the density discontinuing from residue 683, leaving all but the first 15 residues of DV unresolved. Potentially, because of the high flexibility, the rest of DV as well as the tips of the fusion loops in DI are not resolved, which is also seen in the prefusion-stabilized ectodomain of HCMV gB[17]. This high flexibility is also notable in the two-dimensional classes of the prefusion conformation side views that show no clearly defined density for the membrane disc (Extended Data Fig. 8b). Even the resolved part of the ectodomain shows considerable flexibility within and between protomers, as evident in variability analysis showing a lateral translation of DII and αC of DIII by roughly 5 Å (Supplementary Video 1). In a recent preprint describing a prefusion-stabilized ectodomain of HSV-1 gB a similar flexibility of the central helix was seen, with two structures showing the helices either apart (open form) or in close proximity at the centre (closed form)[43]. Therefore, binding of the nanobody does not completely fix gB in a single position as in the case of the stabilizing mutations, as areas such as the transmembrane region remain flexible, but rather seems to prevent the principal conformational changes required for fusion. Some of the prefusion-specific features resolved in our mutationally stabilized

structure are also found in wild-type gB (Supplementary Fig. 2b). The N-terminal helix covering residues 90–106 shows similar interactions with DII as in HSV-1 gB (Fig. 5c). Probably owing to the high flexibility of DII, some of its features are not well resolved. Still, the density is clear enough to build the first ten residues of the αX helix, showing the same upright arrangement as in the stabilized structure (Figs. 2c and 5c). Overlay of the stabilized with the wild-type structure showed only minor differences in the regions where stabilizing mutations were introduced (Fig. 5d). Notably, in DIII, the N terminus of the central helix (αC) forms the same loop as seen in the proline-stabilized gB (Figs. 2d and 5c). This observation corroborates our previous finding, that an amino acid change to proline in that region of DIII supported this natural bend and prevented the transition into postfusion conformation and therefore inhibited membrane fusion[16]. Furthermore, it confirms that introduction of our stabilizing mutations in gB led to preservation of the native prefusion conformation.

The interface between the nanobody and HSV-2 gB is as well resolved as in the stabilized structure (Fig. 4 and Extended Data Fig. 3d). No density for the bound nanobody was detected in either two-dimensional or three-dimensional reconstructions of the postfusion conformation.

By contrast, the reconstruction for prefusion gB shows the protein bound exclusively by Nb1_gbHSV, indicating that binding of the nanobody is limited to this conformation and also indicates that it prevents the protein from transitioning into postfusion. The structural model built in the density for the gB–nanobody complex shows a similar interaction network of hydrogen bonds and salt bridges as seen before (Extended Data Table 1 and Extended Data Fig. 8e), confirming the cross-reactivity of Nb1_gbHSV. The fact that prefusion gB is found only with the nanobody-bound form (Fig. 5) implies the prevention of the substantial conformational changes necessary during HSV-1 infection, therefore explaining the neutralizing activity of Nb1_gbHSV (Fig. 1).

## Discussion

The neutralizing nanobody was a rare hit. It occurred only once in about 200 analysed sequences. This could be due to the fact that, in contrast to newly emerging viruses, there is a co-evolution between humans and herpesviruses that has lasted for millions of years. Therefore, adaptation to the challenge of immune evasion probably included minimizing the exposure of vulnerable epitopes on essential proteins. Most of the structural changes during the conformational switch in gB happen between domains, leaving the individual domains as rigid bodies (Supplementary Discussion and Supplementary Figs. 8 and 9). Still, the protein undergoes a marked change in shape with a more than 30% increase in height (Fig. 5). These two conflicting observations can be reconciled by the fact that most of these necessary changes involve DV, which is buried in the prefusion conformation structure, severely limiting the possible target area. Monoclonal antibodies such as SS106 and SS144 that target DV (amino acids 697–725)[20] are therefore able to bind only the postfusion structure or a potential intermediate, again limiting the possible target area in the prefusion conformation. Hence the differences between both structures in terms of antigenic epitopes are minimal, making it more challenging to specifically target the prefusion structure. Furthermore, analysis of non-neutralizing nanobodies showed that they all target the αX helix at the apex of gB (Extended Data Fig. 6e–g), which is readily accessible in the prefusion conformation on the surface of the virus and could therefore act as a decoy for the immune system while vulnerable epitopes are shielded by glycosylation (Extended Data Fig. 7). Binding affinity measurements of these nanobodies (Supplementary Fig. 4) also showed that even a strong binding preference for the prefusion form does not necessarily correlate with efficient neutralization activity (Extended Data Fig. 1). From this it can be deduced that nanobodies must have the following properties to neutralize: (1) binding to prefusion gB, with (2) sufficient affinity and (3) weaker binding to postfusion, resulting in a change in Gibbs free energy that forms an energy barrier that is sufficient to hold gB in prefusion conformation. Presumably, there is a threshold change in Gibbs free energy that is required for an effective neutralizing effect. This finding further underscores the difficulty in generation, and hence finding, neutralizing nanobodies against gB, and emphasizes the uniqueness of Nb1_gbHSV in terms of binding affinity and epitope specificity. This assessment is supported by a report of a prefusion-stabilized version of the HCMV gB ectodomain used to immunize mice: notably, the stabilized construct did not produce superior neutralization titres compared with a non-stabilized postfusion construct, whereas general complement-independent elicitation of neutralizing antibodies was low[17]. In light of our determined structure, other than for HCMV[4], HSV-1 gB features further differences (Fig. 2) that could be targeted by antibodies, specifically at the apex of the structure where the loop region of αC of DIII is located and at the fusion loops near the membrane. The apex region of gB is decorated with several glycosylation sites (Fig. 4a), notably at N141, which is well positioned to shield the loop region of αC. The changes we have seen in DI would also make for an excellent target, but the close proximity to the MPR (Fig. 3), the inherent flexibility (Fig. 5) and the crowded surface on the virus probably hinder accessibility.

In other class III fusion proteins, structural differences have been shown to allow neutralization by antibodies specific to the prefusion conformation. In rabies virus G protein, two modalities of neutralization were described: (1) targeting an epitope on a single domain that is fully accessible only in prefusion is enough to lock the protein in prefusion conformation, hence inhibiting the fusion process[44] and (2) recognition of an epitope that spans more than three domains, including one on an adjacent protomer, prevents the conformational changes required for fusion[45]. Similarly, the epitope bound by Nb1_gbHSV opens an attractive avenue for neutralization by binding conformational epitopes that span several domains that are close in prefusion but apart in postfusion conformation. Nb1_gbHSV contacts DIV, DI and DIII at the crevice between adjacent protomers (Fig. 4). Given the narrow space in this region, it is possible that conventional antibodies may be too large to access this epitope, especially on the viral envelope as shown for the Fab of antibody SS55, which has a higher neutralizing activity than the full immunoglobulin-γ[21]. Conversely, it remains possible that some antibody paratopes could neutralize the virus by interfering with the structural transition of gB and/or interaction with its neighbouring molecules due to the steric hindrance, which is much greater than that imposed by a nanobody.

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

## Methods

### Alpaca immunization, generation of the nanobody library and phage display selection

gB-presenting vesicles were used for alpaca immunization. Generation of the nanobody library and phage display selection were performed essentially as described previously[40] using biotinylated gB vesicles as baits. Recovered clones were identified by sequencing, and representatives of all classes were expressed as described below.

### Cytoplasmic bacterial expression and purification of VHH antibodies

VHH antibodies were produced as His14–ScSUMO fusions by cytoplasmic expression in *E. coli* SHuffle Express (New England Biolabs), which allows for formation of disulfide bonds in the cytoplasm. Cells were grown in 300 ml Terrific Broth, and protein expression was induced at 21 °C with 0.08 mM isopropyl β-ᴅ-1-thiogalactopyranoside. Five hours after induction, 5 mM EDTA was added to the culture medium. The bacteria were then pelleted, resuspended in 50 mM Tris-HCl pH 7.5, 20 mM imidazole, 300 mM NaCl (lysis buffer) and frozen in liquid nitrogen. Cells were lysed by thawing and sonication, and insoluble material was removed by ultracentrifugation at around 160,000$g$ (approximately 1 h, T647.5 rotor; Thermo Fisher Scientific).

Soluble material was applied to a 1 ml Ni$^{2+}$ chelate column. The matrix was washed with lysis buffer, with further washing steps including 0.2% (w/v) Triton X-100 or 1 M NaCl, respectively. The VHH antibody was eluted by cleaving the His14–SUMO-tag using 100 nM *Saccharomyces cerevisiae* Ulp1p for 2 h at room temperature. The eluted VHH antibodies were frozen in liquid nitrogen and stored at −70 °C until further use.

### Labelling of nanobodies

Nanobodies were labelled with an engineered C-terminal cysteine using maleimides of Alexa Fluor 488 or Alexa Fluor 647 for immunofluorescence as described previously[40].

### Mammalian cell lines

All mammalian cell lines were acquired originally from the American Type Culture Collection (ATCC). Authenticity of cell lines was confirmed through morphology and growth behaviour in cell-type-specific media. All cell lines were tested regularly for mycoplasma contamination and all results were negative. No commonly misidentified cell lines (according ICLAC and NCBI Biosample) were used in this study.

### Neutralization tests and IC$_{50}$ determination

To test for neutralization of the nanobody, plaque reduction assays were performed. Vero cells (ATCC, CCL-81) were seeded in 24-well plates and grown overnight to form a confluent cell layer. Before cell layer infection using around 20 plaque-forming units per well (160 plaque-forming units ml$^{-1}$), viruses were pre-incubated for 30 min at room temperature with concentrations ranging from 1 nM to 1 μM of nanobodies. Subsequently, infection was performed for 1.5 h at room temperature before inoculum was removed and cells were washed with PBS. After buffer removal, cells were overlaid with 1.2% avicell in Dulbecco's Modified Eagle Medium and 5% fetal bovine serum (FBS) to prevent secondary infection through the cell medium and to promote cell-to-cell spread (plaque formation). At 48–72 h after infection, cells were fixed with 4% paraformaldehyde (PFA) in water and stained with 0.1% crystal violet in 2% ethanol. Plaques were counted manually. IC$_{50}$ values were calculated using the AAT Bioquest calculator (https://www.aatbio.com/tools/ic50-calculator).

### Vesicle preparation

Vesicles were prepared as described[26]. In brief, BHK-21 (C13) (ATCC, CCL-10) cells were grown in Glasgow minimal essential medium (GMEM) supplemented with 20 mM HEPES pH 7.4, 2% (v/v) tryptose phosphate broth and 2% (v/v) FBS. At around 70% confluency, cells were transfected transiently with expression plasmids encoding wild-type gB (Uniprot, A1Z0P7), and derivatives thereof featuring C-terminal tags and/or mutations. Cells were grown for a further 48 h with a medium exchange to serum-free GMEM after 24 h. Vesicles were collected from the supernatant by differential centrifugation and resuspended in 20 mM HEPES pH 8, 150 mM NaCl. For alpaca immunizations, vesicles were resuspended in 20 mM HEPES pH 8, 150 mM NaCl and 250 mM sorbitol.

### Expression tests

Protein expression of different double cysteine mutants was determined by testing for cell surface localization of gB. BHK-21 cells were grown in Ibidi μ-Slide eight-well chambered coverslips for 24 h before transient transfection using Lipofectamine 2000. Cells were transfected with plasmids for expression of wild-type HSV-1 gB or double cysteine mutants. HSV-1 gB (1–852), unable to localize to the surface, was used as negative control. At 24 h post transfection, cells were stained with 0.5 μM of Alexa 647 labelled Nb1_gbHSV in GMEM for 1 h at 37 °C. Next, cells were washed with PBS before staining for 10 min with 1 μg ml$^{-1}$ Hoechst stain. After two PBS washes, cells were fixed with 4% PFA and imaged subsequently in PBS on a Leica DMi8 system at ×40 magnification. Surface staining efficiency was determined by counting nuclei and Alexa 647-stained cells in five random images per well. The percentage of stained cells relative to the percentage for wild-type gB was calculated for three independent experiments.

Protein expression of different gB mutants in vesicles and whole-cell extracts was tested as described in ref. 16 using SDS–PAGE and western blotting with a 1:5,000 dilution of rabbit anti-His6 antibody (Abcam) followed by a 1:5,000 dilution of anti-rabbit-HRP (Sigma-Aldrich Chemie GmbH). Twin-Strep tagged proteins were detected using a 1:4,000 dilution of Strep-Tactin HRP (iba Lifesciences). As a loading control, western blots were re-probed using a 1:2,500 dilution of mouse anti-GAPDH antibody (Sigma-Aldrich Chemie GmbH) followed by a 1:5,000 dilution of anti-mouse-HRP (Sigma-Aldrich Chemie GmbH); uncropped blots are provided in Supplementary Fig. 10.

### Cross-reactivity tests

BHK-21 cells were grown in Ibidi μ-Slide eight-well chambered coverslips for 24 h before transient transfection using Lipofectamine 2000. Cells were transfected with plasmids for expression of different gB homologues, namely HSV-1, HSV-2, VZV, HCMV or EBV. In addition, as negative control, the structurally related glycoprotein G from vesicular stomatitis virus was transfected. All constructs are based on the respective wild-type sequence, and encode a C-terminally added fluorescent sfEGFP tag. An amount of 1 μg DNA + 1 μl lipofectamine per well in 200 μl GMEM was used for transfection. At 24 h post transfection, cells were washed with PBS before addition of 200 μl fresh GMEM including 0.5 μM of Alexa 647 labelled Nb1_gbHSV nanobody. After 1 h incubation at 37 °C, cells were washed with PBS and fixed subsequently in 4% PFA for 10 min. Cells were kept in PBS for imaging. Images were analysed using FIJI software[48] and the JaCoP plugin[49] was used to calculate Pearson's correlation coefficients of the measured fluorescent signal in the green and red channel.

### Protein purification and reconstitution

HEK293T cells (ATCC, CRL-3216) were transduced stably as described in ref. 31 with mutationally prefusion-stabilized HSV-1 gB or wild-type HSV-2 gB featuring a C-terminal Twin-Strep-tag. For co-expression, cells were previously transduced with a construct for Nb1_gbHSV featuring a signal peptide derived from HSV-1 glycoprotein H and a C-terminal His$_6$ tag. To test for transduction efficiency, a separate reading frame on the same vector behind an internal ribosome entry site

encodes for emerald green fluorescent protein. Expression vectors were obtained from Addgene (113901 and 113888). Stable cells were grown for 3–5 days in F12 medium (Gibco) with 5% FBS after reaching confluency before detachment in cold PBS. Cells were pelleted and snap-frozen in liquid nitrogen. Frozen pellets were resuspended in 20 mM HEPES, 500 mM NaCl and 50% glycerol using a douncer. A final concentration of 1.5% dodecyl maltoside and 0.15% cholesteryl hemisuccinate (Anatrace) was used for solubilization. After supernatant clearance, affinity purification was done using Strep-Tactin XT 4Flow high capacity resin (IBA). Reconstitution in peptidiscs[32] (PEPTI-DISC BIOTECH) was achieved by dialysis of detergent-purified protein with peptide in the presence of Bio-Bead SM-2 resin (Bio-Rad). Size exclusion chromatography (SEC) was done on a customized AEKTA Pure (Cytiva) in 20 mM HEPES, 300 mM NaCl, 5 mM arginine, 5 mM glutamate using a Superose 6 Inc. 3.2/300 column. Peak fractions were pooled and concentrated using Amicon Ultra 0.5 ml 100 kDa molecular weight cut-off spin concentrators. For nanobody co-expressed gB, solubilization and washing steps were done in the presence of 1 μM Nb1_gbHSV. The IgV domain of PILRa was expressed with an N-terminal His$_{14}$-NEDD8 tag in BL21DE Rosetta2 cells using autoinduction in Terrific Broth medium overnight at 25 °C. After pelleting at 10,000$g$ for 20 min, cells were resuspended in buffer A (50 mM HEPES pH 7.5, 500 mM NaCl, 30 mM imidazole) lysed in an LM10 Microfluidizer (Microfluidics) before spinning at 30,000$g$ for 30 min. Supernatant was incubated with NiNTA Sepharose HP (Cytiva) for 2 h and applied to a column for washing steps with buffer A using more than 25× the bead volume. Bound protein was eluted with 2.5× the bead volume using 500 mM imidazole in buffer A. After addition of 1 μM final concentration of bdNEDP1, the mixture was dialysed overnight against buffer A. Dialysed eluate was incubated with the same volume of NiNTA Sepharose HP for 2 h before flowthrough was collected through a column and concentrated before SEC using a Superdex 200 Inc. 10/300 column in 20 mM HEPES pH 7.8 and 300 mM NaCl. Peak fractions were pooled and snap-frozen to be stored in −70 °C before use. Uncropped gels are provided in Supplementary Fig. 11.

## Cryo-EM methods

Purified gB protein (3.5 μl; roughly 2.5 μM) or a 1:20 mixture of gB with Nb1_gbHSV were added to glow discharged Quantifoil R2.1 copper grids before a 3-s blotting step followed by plunging into a liquid ethane/propane mixture using a Thermo Fisher Scientific Vitrobot Mark IV. Frozen grids were imaged using a Titan Krios microscope (Thermo Fisher Scientific) operated at 300 kV and equipped with an X-FEG electron source and a K3 direct electron detector (Gatan) with a Bioquantum post-column energy filter operated in zero-loss imaging mode with a 20 eV slit width. Data were collected in fringe-free imaging aberration-free image shift mode using SerialEM[50]. Details for individual datasets are summarized in Extended Data Table 2.

For protein conformation analysis on vesicles, 3.5 μl gB vesicles were mixed with 0.5 μl 5 nm nanogold and were plunged as described before for cryo-ET and single-particle analysis. Tomograms were acquired, processed and reconstructed as described previously[16].

## Data processing and structure determination

Particle picking was performed in WARP using a custom trained template[51]. Particle positions were imported in CryoSPARC[52] for particle extraction and further processing. All subsequent processing steps were performed in CryoSPARC. An initial model was built using ModelAngelo[53] and missing residues were added in Coot[54]. Initial refinement was done with Isolde[55]. In CCP-EM Doppio (https://www.ccpem.ac.uk/), the model was fitted in the map using MolRep[56] and the structure was refined using Refmac Servalcat[57] and Phenix[58]. Where density permitted, N-acetyl-glucosamine sugars were added to glycosylation sites using ChimeraX. These glycosylated complexes were finally refined with TEMPy-ReFF v.1.2 (ref. 59). Validation was

done using MolProbity v.4.5.2 (ref. 60), FDR backbone[61] and Find-MySequence[62]. Interacting residues and surfaces were determined using the 'protein interfaces, surfaces and assemblies' service PISA at the European Bioinformatics Institute[63] (http://www.ebi.ac.uk/pdbe/prot_int/pistart.html) and ChimeraX[64]. Root-mean-square deviation between postfusion HSV-1 and HSV-2 gB was calculated in ChimeraX using PDB 2GUM (ref. 33). Multiple sequence alignment for epitope conservation was performed using Clustal Omega[65] and conservation score was calculated as described in refs. 66,67 using SnapGene v.8.1. Figures were prepared using ChimeraX v.1.8 and Illustrator/Photoshop 2024 (Adobe). Manders' Overlap Coefficient analysis to evaluate the goodness of model to map fit was done using TEMPy2 (ref. 68). Structure alignments were done by computing secondary structure elements using DSSP[69], and plotted against a sequence alignment of HSV-1 and HSV-2 gB using the uniprot alignment tool (Clustal Omega GUI[70]).

## Analysis of gB conformation on vesicles

Acquired videos were imported into CryoSPARC[52] for patch motion correction and export for statistical analysis. Identified vesicles were sorted into three classes on the basis of their gB conformations: prefusion only, postfusion only and mixed population. Using the total amount of counted vesicles per gB construct, the percentages of each class were calculated using Excel v.16.16.27 (Microsoft).

## Nanobody epitope mapping

Cryo-EM sample preparation was done as described earlier. A 2:1 molar ratio mixture of peptidisc reconstituted prefusion-stabilized and wild-type gB with a final concentration of 0.55 μM was prepared and used for all datasets. Different nanobodies or buffer were added in 50× molar excess and incubated for 5–10 min before plunge freezing. Data acquisition and processing was performed as described earlier. Details for individual datasets are summarized in Extended Data Table 2. Particles of the two conformations were separated in two-dimensional classification and further refined by heterogeneous refinement with individually generated ab initio models. Final densities, used for difference mapping, were produced by non-uniform refinement. Difference maps between the nanobody containing samples and the apo structures were produced in ChimeraX and fitted onto the respective apo structure of gB.

## Affinity measurements

Binding kinetics of nanobodies with prefusion or postfusion gB were measured using grating-coupled interferometry on a Creoptix WAVE-delta instrument (Creoptix AG)[71]. All measurements were performed using quasi-planar polycarboxylate polymer WAVEchips with four channels (Creoptix AG) at 25 °C. Chips were coated with Strep-Tactin XT by amine coupling using a Twin-Strep-tag capturing kit (IBA Lifesciences). Prefusion-stabilized gB or postfusion gB (ectodomain of wild type) was immobilized on separate channels on the WAVEchip using a C-terminal Twin-Strep-tag to a density of 400 pg mm$^{-2}$. During gB immobilization and kinetics measurements, 20 mM HEPES, pH 8.0, 300 mM NaCl was used as running buffer. Kinetics measurements were conducted using the waveRAPID mode with the tight binder protocol with 200 nM for Nb1_gbHSV and 500 nM for Nb2_gbHSV, Nb3_gbHSV and Fab SS55 (control). After each injection of nanobody or Fab, the chip was regenerated using 3 M GuHCl (IBA Lifesciences) before re-immobilization of fresh gB protein in the respective channels. For measurement of Nb4_gbHSV and PILRα (control) binding kinetics, the waveRAPID intermediate binder protocol was used, without regeneration or further gB immobilization. Nb4_gbHSV and PILRα were used at a concentration of 1 μM. All measurements were performed three times and evaluated using the WAVEcontrol software and double referenced against the reference channel (Strep-Tactin XT coated) and a blank. Every tight binder measurement

was fitted using traditional fitting and an end crop was set to 1,000 s. For intermediate binder measurements, the end crop was kept at default.

## Microscale thermophoresis

To determine the binding affinity of Nb1_gbHSV to prefusion-stabilized gB, covalently labelled Nb1_gbHSV-Alexa 647 (target) was diluted to a concentration of 400 pM in 20 mM HEPES, pH 8.0, 300 mM NaCl, 0.005% Tween20 (assay buffer). Prefusion-stabilized gB (ligand) was diluted to 20 nM in assay buffer and ligand buffer (20 mM HEPES, pH 8.0, 300 mM NaCl) was diluted 1:138 in assay buffer. Then, a 1:1 dilution series of gB with 16 dilutions was prepared in the diluted ligand buffer. Finally, the gB dilutions were mixed with equal volumes of the diluted Nb1_gbHSV-Alexa 647 to final gB concentrations ranging from 10 nM to 0.3 pM. The final concentration of Nb1_gbHSV in the assay was 200 pM. The Nb1_gbHSV-Alexa 647 gB mixture was incubated for around 24 h at 4 °C in the dark. Samples were loaded into standard Monolith capillaries and measured using the Monolith 2020 pico-RED instrument (NanoTemper) at 100% excitation and medium microscale thermophoresis power at 25 °C. The data were evaluated using the MO.Control v.2 software (NanoTemper). Outliers were excluded from the data before fitting to determine binding affinities. The measurements were repeated three times.

## Animal work

One female alpaca kept at the Alpaca Facility of the Max Planck Institute for Multidisciplinary Sciences (Göttingen) was immunized using gB-presenting vesicles. The alpaca project (immunizations and blood sampling) has been approved by the animal welfare authority LAVES with the reference numbers 33.9-42502-05-13A351, 33.9-42502-05-17A220 and 33.19-42502-04-22-00210.

## Statistical analyses

To calculate the $IC_{50}$ of Nb1_gbHSV in Fig. 1b, response curves were fitted using the equation:

$$y = Min + (Max - Min)/1 + (x/IC_{50})^{Hill\ coefficient}$$

using the AAT Bioquest calculator (https://www.aatbio.com/tools/ic50-calculator).

For Extended Data Fig. 1, averages, s.d. and s.e.m. were calculated using Excel v.16.16.27(201012) (Microsoft).

For Extended Data Fig. 6, the Pearson's correlation coefficients of the measured fluorescent signals were calculated using the JaCoP plugin[49] in Fiji[48].

## Material availability

All unique materials used are available upon request from the authors. The use of nanobodies generated in this study might be subject to a material transfer agreement.

## Reporting summary

Further information on research design is available in the Nature Portfolio Reporting Summary linked to this article.

## Data availability

The electron microscopy density maps have been deposited in the Electron Microscopy Data Bank under EMD-52963, EMD-52965, EMD-52863 and EMD-52966. The corresponding models have been deposited with the PDB under 9Q9L, 9Q9N, 9IH8 and 9Q9S. In preparation of this manuscript the following atomic structures have been used that are already available in the PDB: 1IGT, 6CGR, 6Z9M, 5V2S, 2GUM, 7KDP, 8VG6 and 3DUZ. For sequence alignments the following sequences were used that are already available on the UniProt database:

Q4JR05, P08666, A1Z0P7, P03188, F5HB81, F5HB53, P52352, P36320 and P36319. Source data are provided with this paper.

## Code availability

No custom code or mathematical algorithm was developed in this work.

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

**Acknowledgements** We are grateful for the SS55 Fab fragment and the gB ectodomain construct, which were gifts from G. Cohen and M. Backovic. We also thank L. Schatz, C. Whittle and R. Rosch for experimental support. We thank P. Nissen for the peptidiscs recommendation and T. Zeev-Ben Mordehai for advice on grid preparation. In addition, we thank L. Baker, J. Hellert and M. Backovic for helpful discussions. We also thank U. Teichmann and her team for alpaca care, immunizations and blood samplings. This work was supported by the Leibniz Institute of Virology as part of the Leibniz ScienceCampus InterACt (supported by the BWFGB Hamburg and the Leibniz Association) and a Wellcome Trust Collaborative Award in Science (209250/Z/17/Z). The work was also part of the ASPIRE project supported by the German Federal Ministry of Education and Research (BMBF). H.E. and J.N. are supported by DFG Research training group grants 2771 and 2887, respectively. We thank the Sample Preparation and Characterisation facility at EMBL and at CSSB for assistance (S. Niebling) with affinity measurements and the Advanced Light and Fluorescence Microscopy (ALFM) Facility at CSSB for support with light microscopy image acquisition and analysis. All electron microscopy was performed at the Multi-User CryoEM Facility at the Centre for Structural Systems Biology, Hamburg, supported by the University of Hamburg and DFG grant numbers INST 152/772-1|152/774-1|152/776-1 FUGG.

**Author contributions** B.V. and K.G. conceptualized the project. B.V. designed and performed experiments, virus neutralization, protein purification and sample preparation, electron microscopy data acquisition, solved and analysed the structures, and wrote the manuscript. H.E. performed immunofluorescence, protein expression and purification, electron microscopy sample preparation and data acquisition, and fluorescence and electron microscopy image analysis. J.N. conducted binding affinity measurements and electron microscopy data acquisition. T.M. and M.T. analysed structures, modelled glycosylations and performed structure refinements. D.G. conceptualized, oversaw and performed experiments for nanobody generation and purification. R.R. and J.S. performed phage display, production and

labelling of nanobodies and J.K. generated the nanobody library. K.G. oversaw the project and reviewed experimental plans, data interpretation and writing.

**Funding** Open access funding provided by Universität Hamburg.

**Competing interests** B.V., D.G. and K.G. are inventors on a patent application for Nb1_gbHSV and Nb2_gbHSV (application number: EP25159392.7). The other authors declare no competing interests.

**Additional information**
**Correspondence and requests for materials** should be addressed to Dirk Görlich or Kay Grünewald.

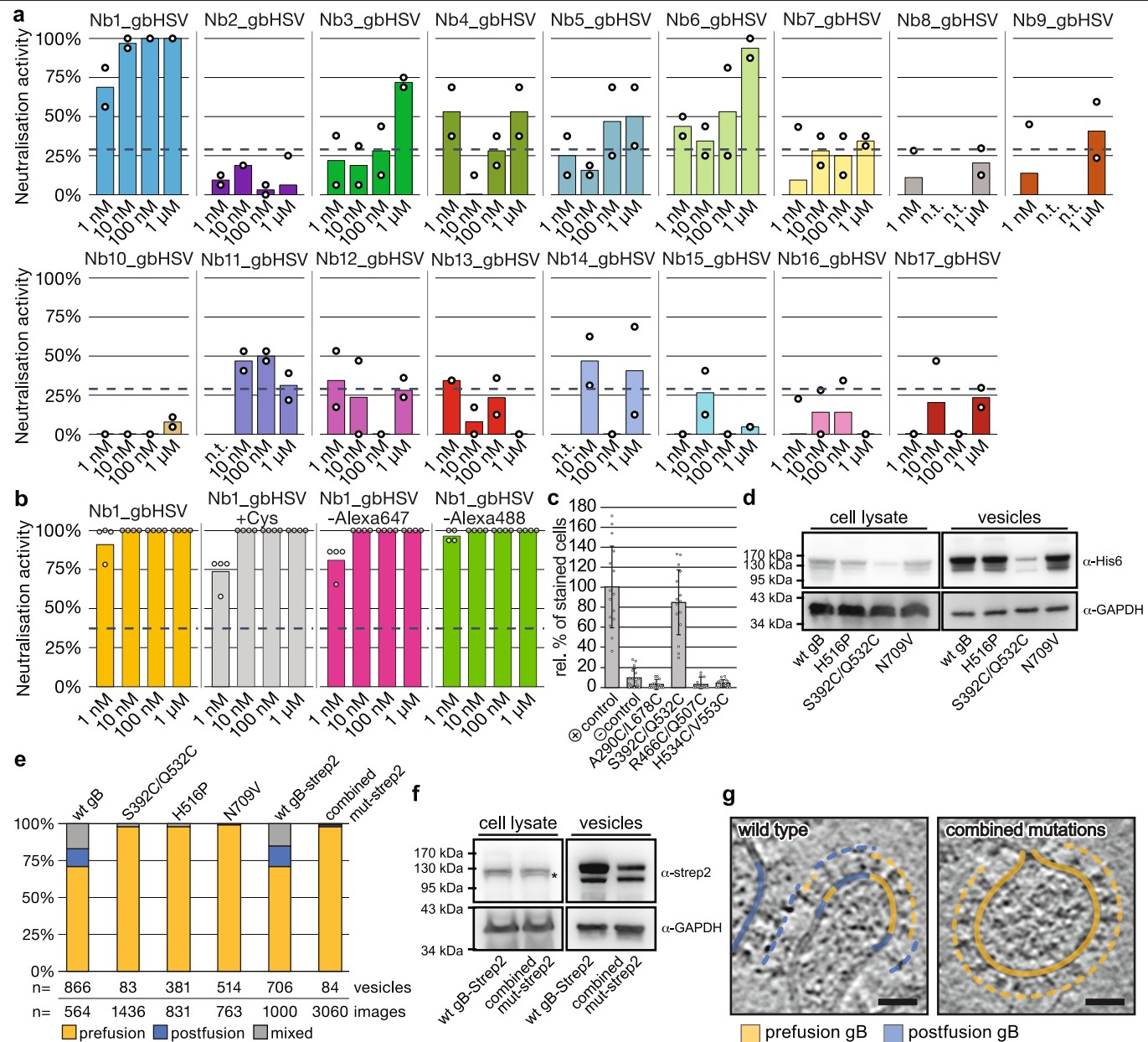

**Extended Data Fig. 1 | Nanobody neutralisation and target stabilization.**
(**a**) Nanobody neutralisation activity tested in HSV-1 plaque reduction assays. Bars represent the average plaque number of n = 2 cell layer infections shown as individual dots and normalised to buffer control in at least two different Nb concentrations. 100% neutralisation: no plaques, 0%: equal or more plaques than the average number of plaques in control. Dashed line shows buffer control range. Infections with more plaques than the control are shown as 0% neutralisation activity. Only Nb1_gbHSV showed high neutralisation activity in all concentrations. n.t. = not tested. (**b**) Neutralisation tests performed and analysed as in (a), but from n = 4 cell layer infections. Neutralisation activity is not influenced by C-terminal cys residue and/or fluorescent labels. (**c**) Expression test by Nb1_gbHSV-Alexa647 staining of cells transfected with indicated double cys mutants. (+)control: wild type gB, (−)control 'fusion dead' construct gB(aa1-852). Bars represent the average percentage of Alexa647-stained of all Hoechst-stained cells relative to the average percentage of wild type gB with individual measurements shown as dots. Error bars: SD of n = 3 independent experiments. Only S392C/Q532C showed consistent cell surface staining similar to wild type gB. (**d**) Western blot analysis of construct

expression in cell lysate and vesicles detected by their C-terminal His$_6$ tag. GAPDH, detected on the same blot, was used as loading control. All constructs express and form vesicles similar to wild type gB, except for S392C/Q532C. Despite the reduced expression, a vesicle band is still detected. Shown is one representative blot of n = 2 successful, independent replicates. For gel source data, see Supplementary Fig. S10a. (**e**) Vesicles produced with wild type gB or indicated mutant were analysed in transmission images and counted in categories depending on gB conformation on the surface. Number of found vesicles roughly aligns with respective expression yields. (**f**) Analysis as in **d**, except gB detection via C-terminal Twin-Strep-tag using Strep-Tactin-HRP. The stabilised construct shows a lower vesicle yield. Shown is one representative blot of n = 2 successful, independent replicates. Unspecific band originating from αGAPDH antibody is marked with an asterisk. For gel source data, see Supplementary Fig. S10b. (**g**) Representative cryoET slices of gB vesicles from one of at least n = 2 independent, successful productions and purifications of vesicles with either wild type or the final expression construct featuring a C-terminal Twin-Strep-tag. Membrane regions are coloured according to the conformation. Bars: 25 nm.

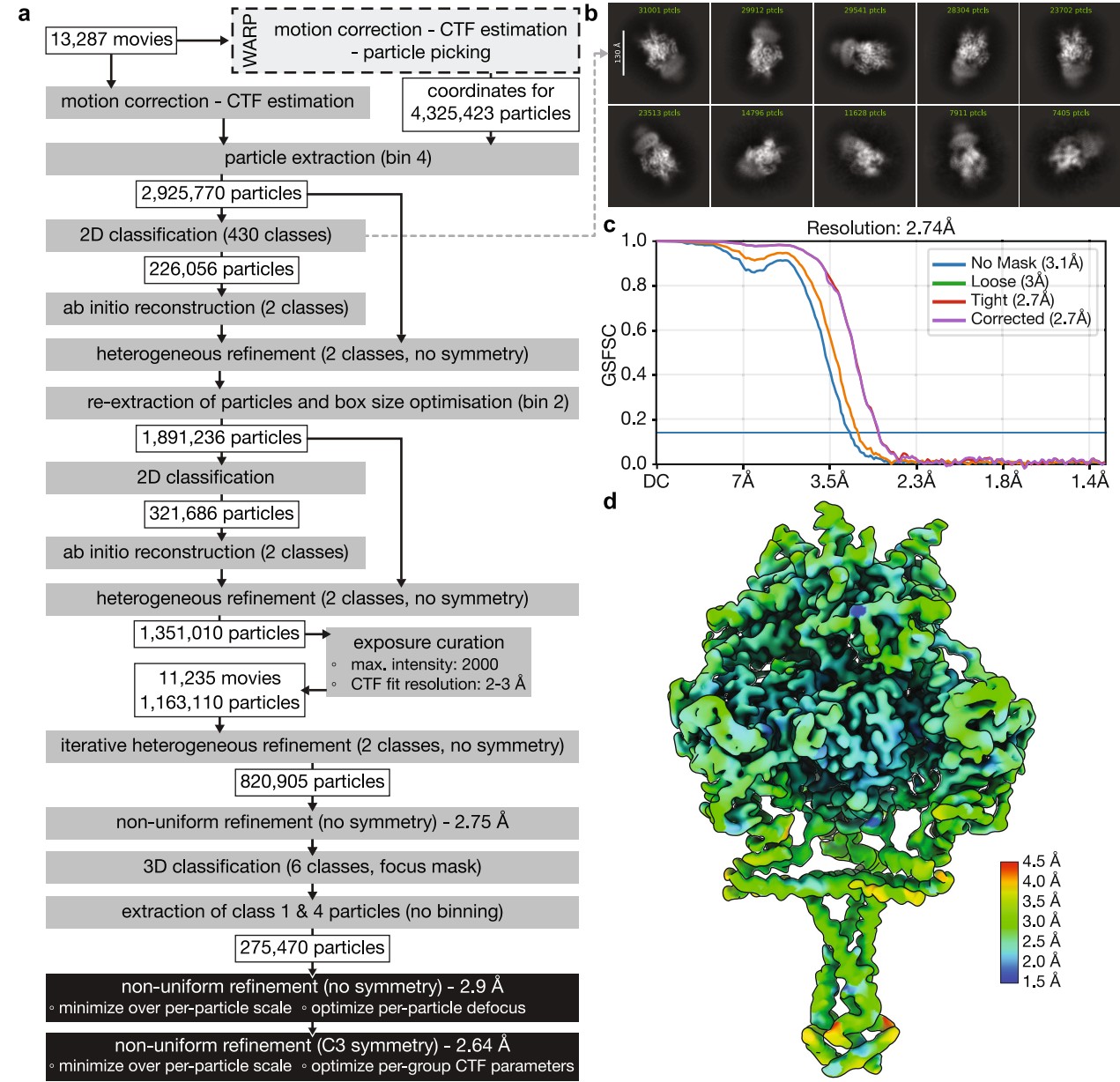

**Extended Data Fig. 2 | The full-length prefusion structure of gB.** (**a**) Processing pipeline, outlining individual steps with particle numbers and options used deviating from default settings. Except initial steps using WARP[51], all steps were done using CryoSPARC[52]. (**b**) 2D classes showing the 10 most populated classes.

(**c**) Fourier shell correlation (FSC) for the density map corresponding to the prefusion structure is shown. Resolution was calculated using an FSC threshold of 0.143. (**d**) Final map coloured by local resolution (FSC threshold 0.143) using a range of 1.5–4.5 Å.

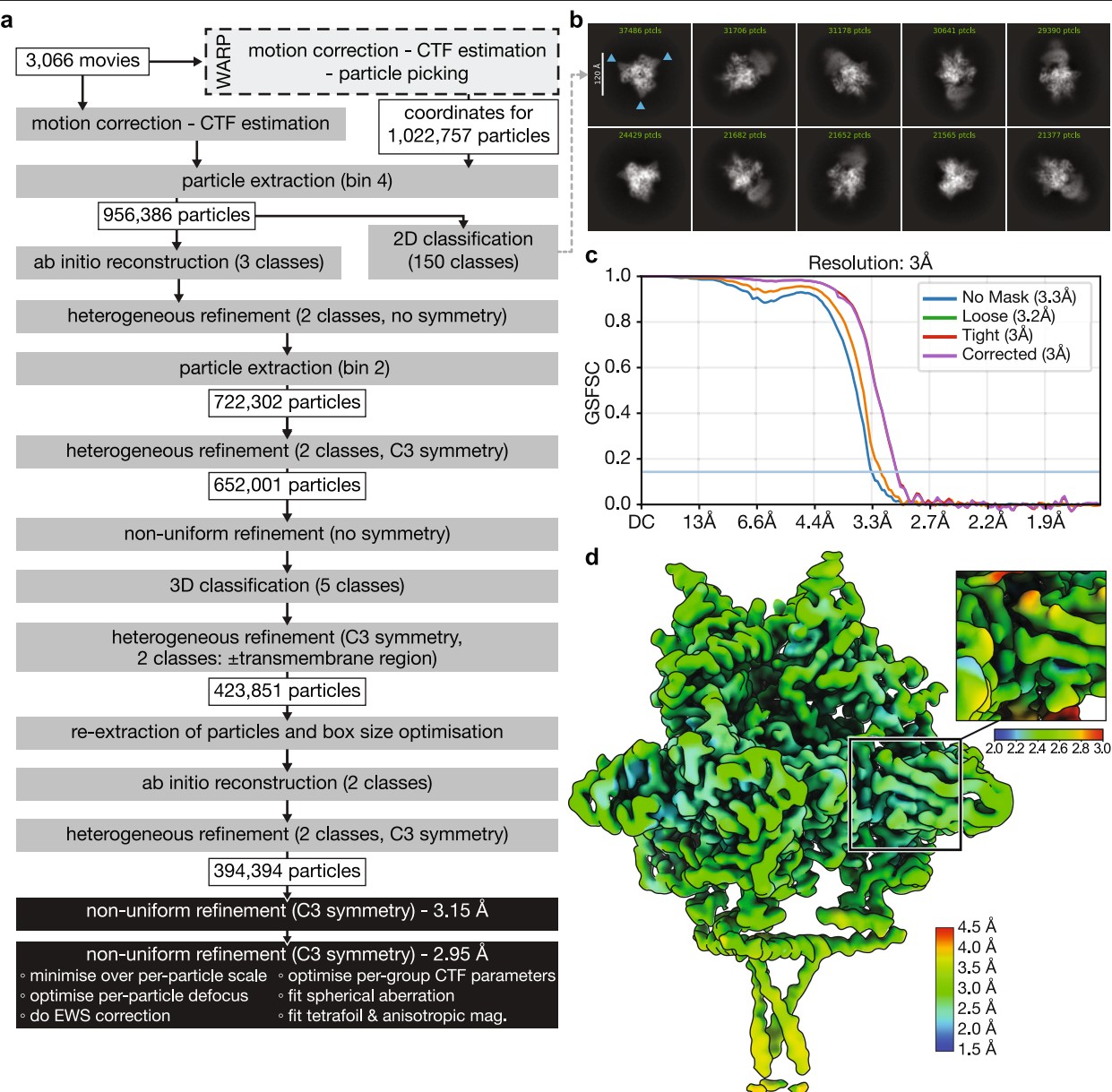

**Extended Data Fig. 3 | Full-length prefusion, Nb1_gbHSV bound structure of gB.** (**a**) Processing pipeline, outlining individual steps with particle numbers and options used deviating from default settings. Except initial steps using WARP[51], all steps were done using CryoSPARC[52]. (**b**) 2D classes showing the 10 most populated classes. (**c**) Fourier shell correlation (FSC) for the density map corresponding to the prefusion structure is shown. Resolution was calculated using an FSC threshold of 0.143. (**d**) Final map coloured by local resolution (FSC threshold 0.143) using a range of 1.5–4.5 Å. The inset shows the local resolution of the nanobody bound region.

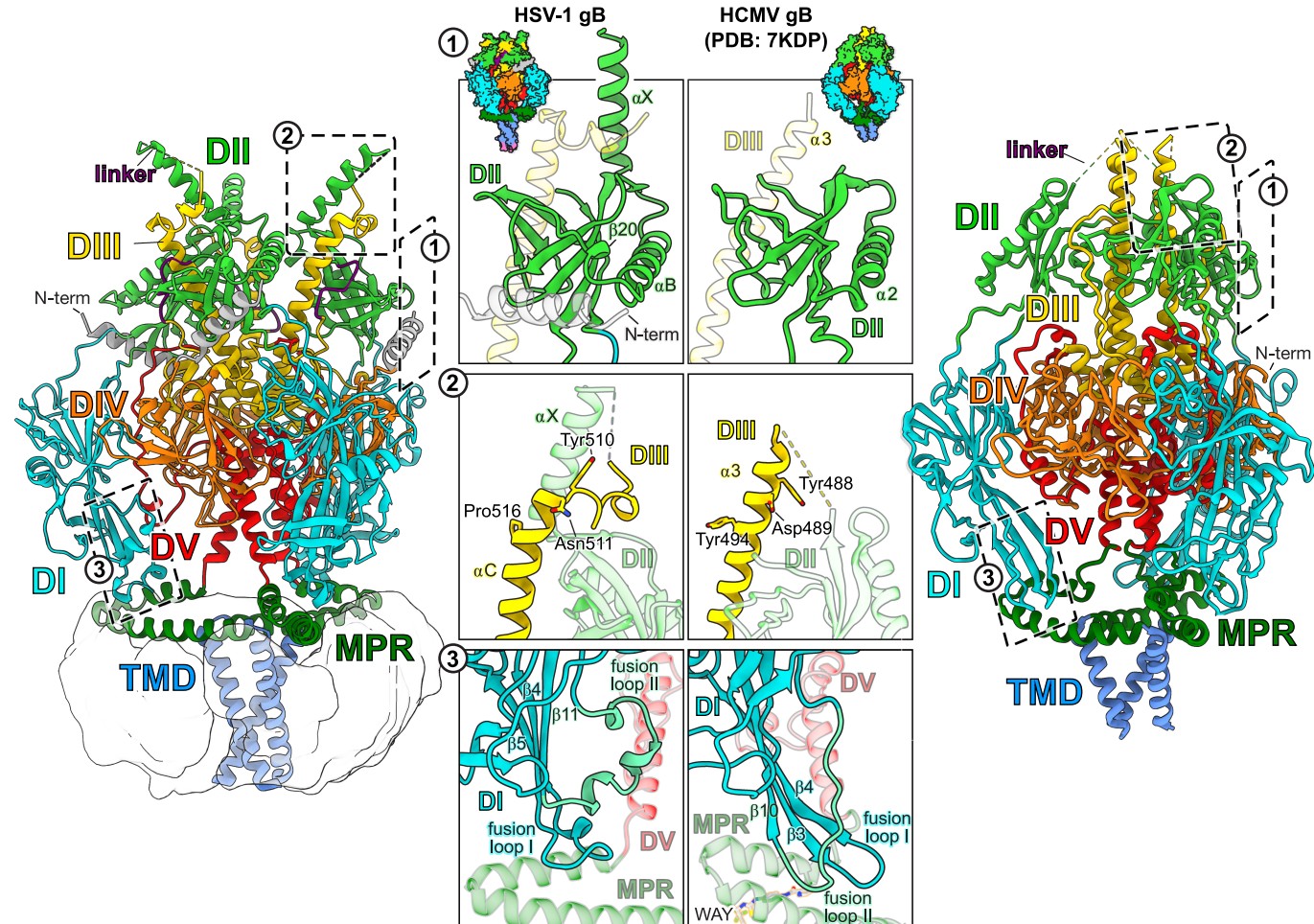

**Extended Data Fig. 4 | Structural differences between HSV-1 and HCMV prefusion gB.** Full gB structures are shown left and right[4] in ribbon style with domains marked by roman numerals and coloured as in[3]. Transmembrane domain (TMD) and MPR are coloured as in[3]. Secondary structure elements are numbered for HSV-1 gB as in[3] and HCMV gB as in[4]. Region and viewing angle of the other panels are indicated by dashed boxes. (1)-(3) Detailed views of the structural features within domains with the HSV-1 gB structure shown in left and HCMV structure (PDB: 7KDP)[4] shown in right boxes indicated by the structural pictograms at the top. The HCMV specific inhibitor WAY-174865 (WAY) is marked in orange. Selected residues are marked in three-letter code and shown in sticks with atoms shown in blue for nitrogen, red for oxygen and yellow for sulphur.

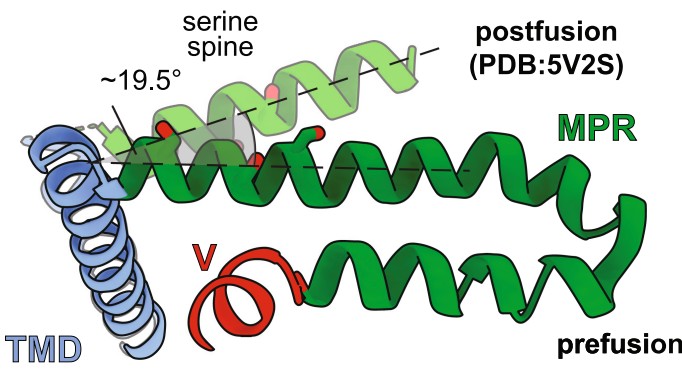

**Extended Data Fig. 5 | MPR change during fusion.** Overlay of ribbon renderings of the MPR and TMD helices of the pre- and postfusion (half transparent PDB: 5V2S)[3], aligned to the TMD showing the indicated angle. The residues of the serine spine are shown in sticks with oxygen atoms shown in red.

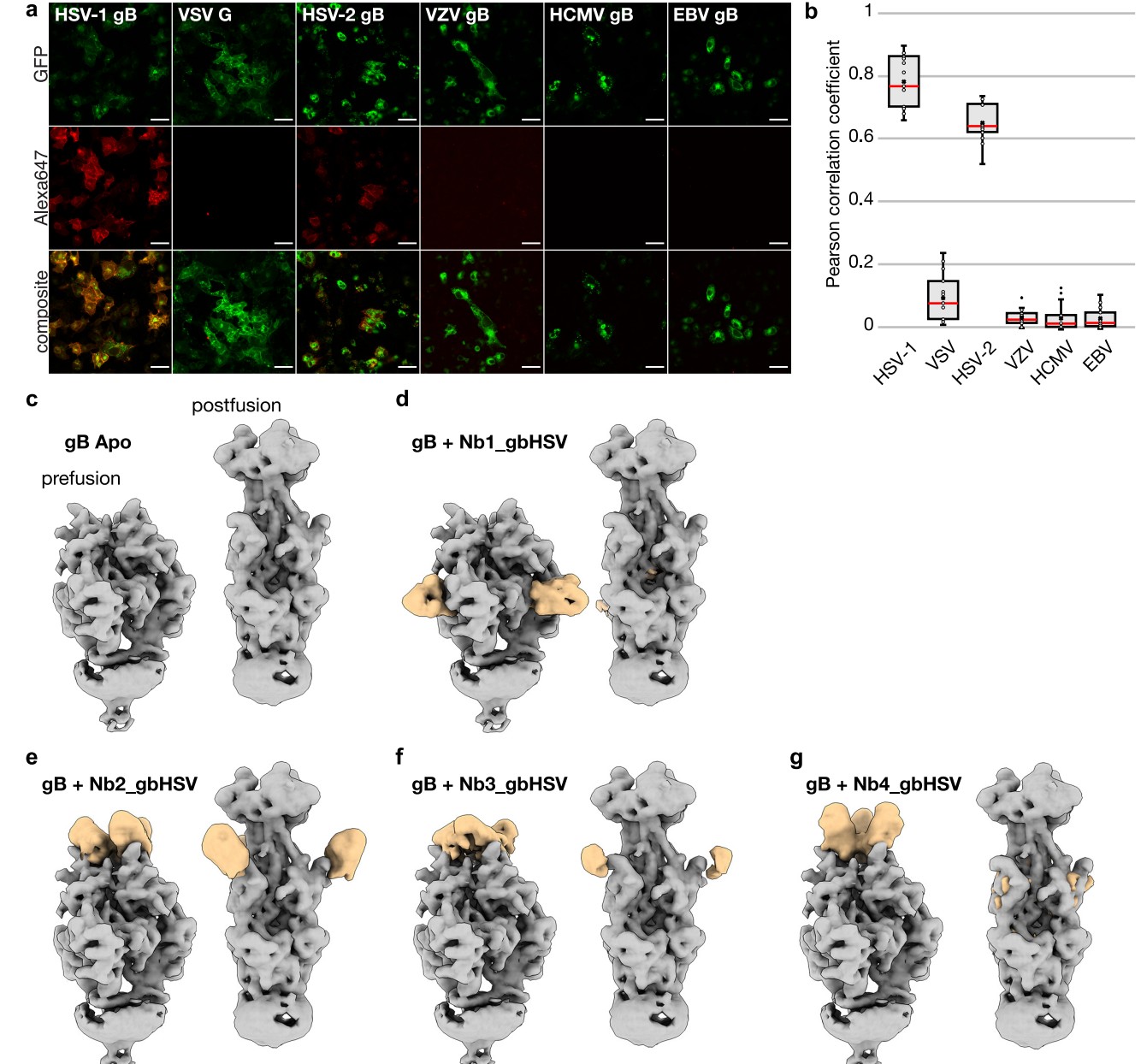

**Extended Data Fig. 6 | Nb1_gbHSV binds gB.** (**a**) Exemplary images of one of at least n = 3 independent, successful replicates of BHK-21 cells transfected with HSV-1, HSV-2, VZV, HCMV or EBV gB; or VSV G featuring a C-terminal sfEGFP tag (green) and stained with Nb1_gbHSV-Alexa647 (red). Bars: 50 μm (**b**) Correlation of fluorescent signal originating from the sfEGFP signal of the gB or VSV-G constructs and the Alexa647 labelled Nb1_gbHSV of three independent experiments (n = 3) in five positions each. Boxes mark the limits of the upper and lower quartile value with whiskers showing the min and max value. Individual data points are marked as white dots while outliers are marked as solid points. The red line and cross mark the median and average, respectively. (**c**) 3D reconstruction of gB in pre- and postfusion conformation, generated from a dataset containing a mixture of both conformations. (**d**) A difference map was generated for each conformation between the Apo structure as shown in (c) and the structure generated from a dataset of a sample containing an additional 50 molar excess of Nb1_gbHSV. The difference map is shown in light brown, mapped onto the Apo structure. (**e**) as in (d), but using Nb2_gbHSV. (**f**) as in (d), but using Nb3_gbHSV. (**g**) as in (d), but using Nb4_gbHSV.

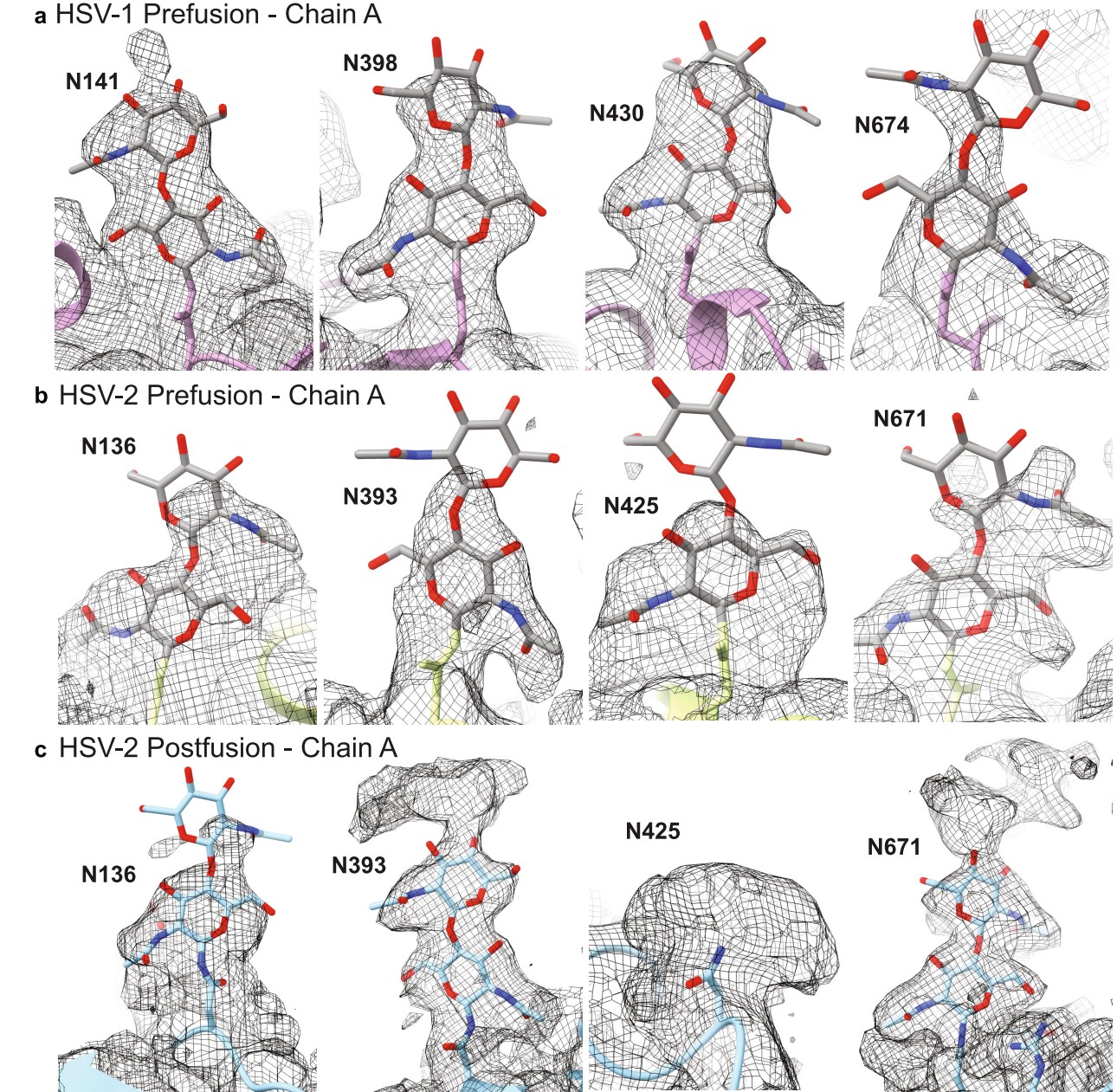

**a** HSV-1 Prefusion - Chain A

N141  N398  N430  N674

**b** HSV-2 Prefusion - Chain A

N136  N393  N425  N671

**c** HSV-2 Postfusion - Chain A

N136  N393  N425  N671

**Extended Data Fig. 7 | Density fits of modelled glycosylations.**
(**a**) N-acetyl-glucosamine sugars were added to glycosylation sites on HSV-1 gB prefusion (Nb1_gbHSV bound) manually using ChimeraX and refined using TEMPy-ReFF v1.2.[59]. Asparagine residues carrying the glycosylation are marked in one-letter code on chain A of gB. (**b**) as in (a) for HSV-2 gB in prefusion conformation. (**c**) as in (a) for HSV-2 gB in postfusion conformation. The density at position N425 did not allow confident fitting of N-acetyl-glucosamine.

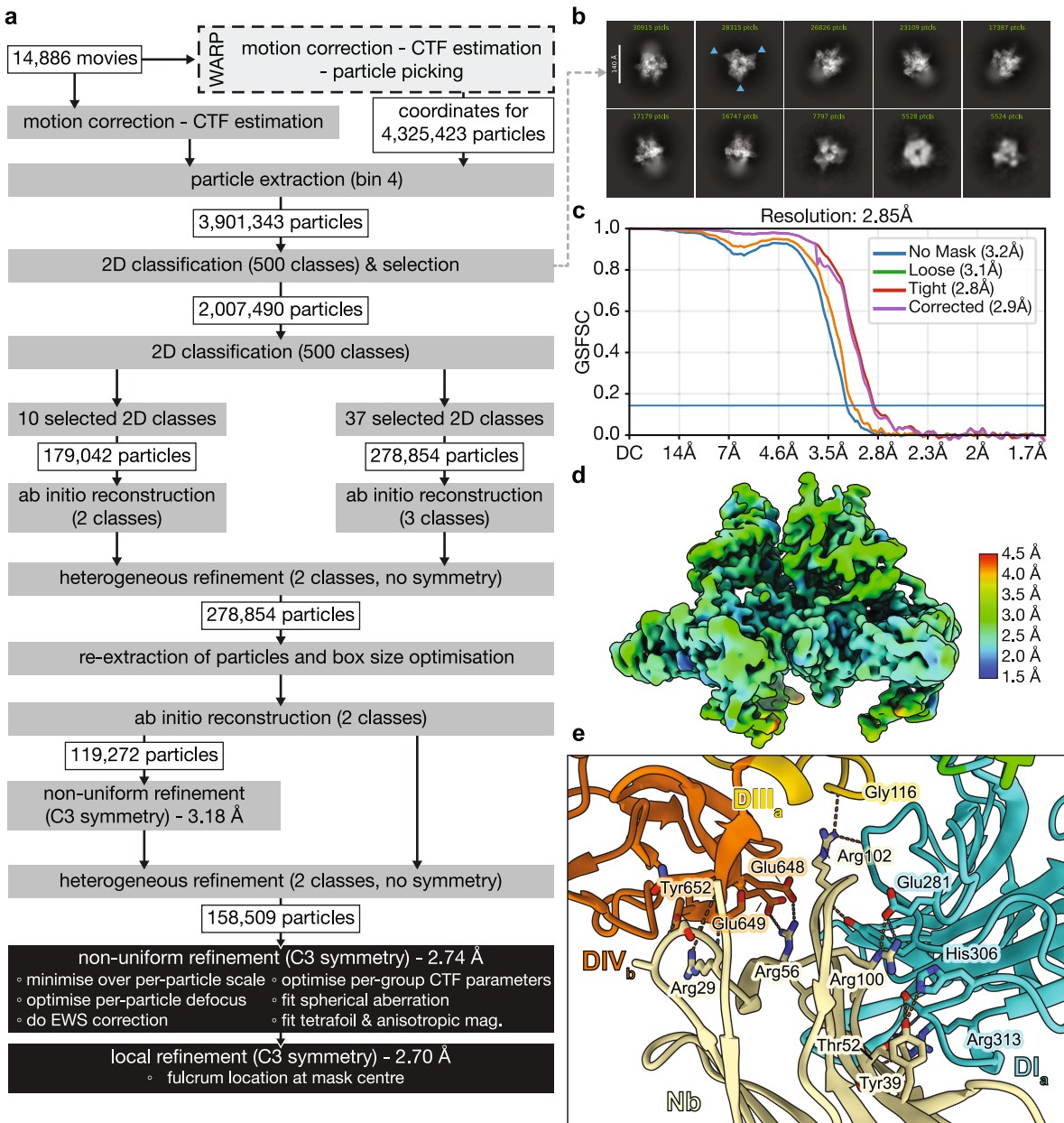

**Extended Data Fig. 8 | HSV-2 gB bound by Nb1_gbHSV. (a)** Processing pipeline, outlining individual steps with particle numbers and options used deviating from default settings. Except initial steps using WARP[51], all steps were done using CryoSPARC[52]. **(b)** 2D classes showing the 10 most populated classes. Blue arrowheads mark the position of Nb1_gbHSV. **(c)** Fourier shell correlation (FSC) for the density map corresponding to the prefusion structure is shown. Resolution was calculated using an FSC threshold of 0.143. **(d)** Final map coloured by local resolution (FSC threshold 0.143) using a range of 1.5–4.5 Å. **(e)** Ribbon rendering of the nanobody bound gB structure, zoomed in on the interaction between Nb1_gbHSV (Nb) and DIV, DI & DIII, DI. Domain numbers are shown in roman numbering. Hydrogen bonds are marked by blue dashed lines with a distance and angle tolerance of 0.400 Å and 20° respectively. Bonds not meeting the precise criteria are shown in orange dotted lines. and amino acid names are given in 3 letter code with the according position. Interacting residues are shown in sticks with atoms shown in blue for nitrogen, red for oxygen and white for hydrogen. Protomers are indicated by lower case letters a-c on domain numbers.

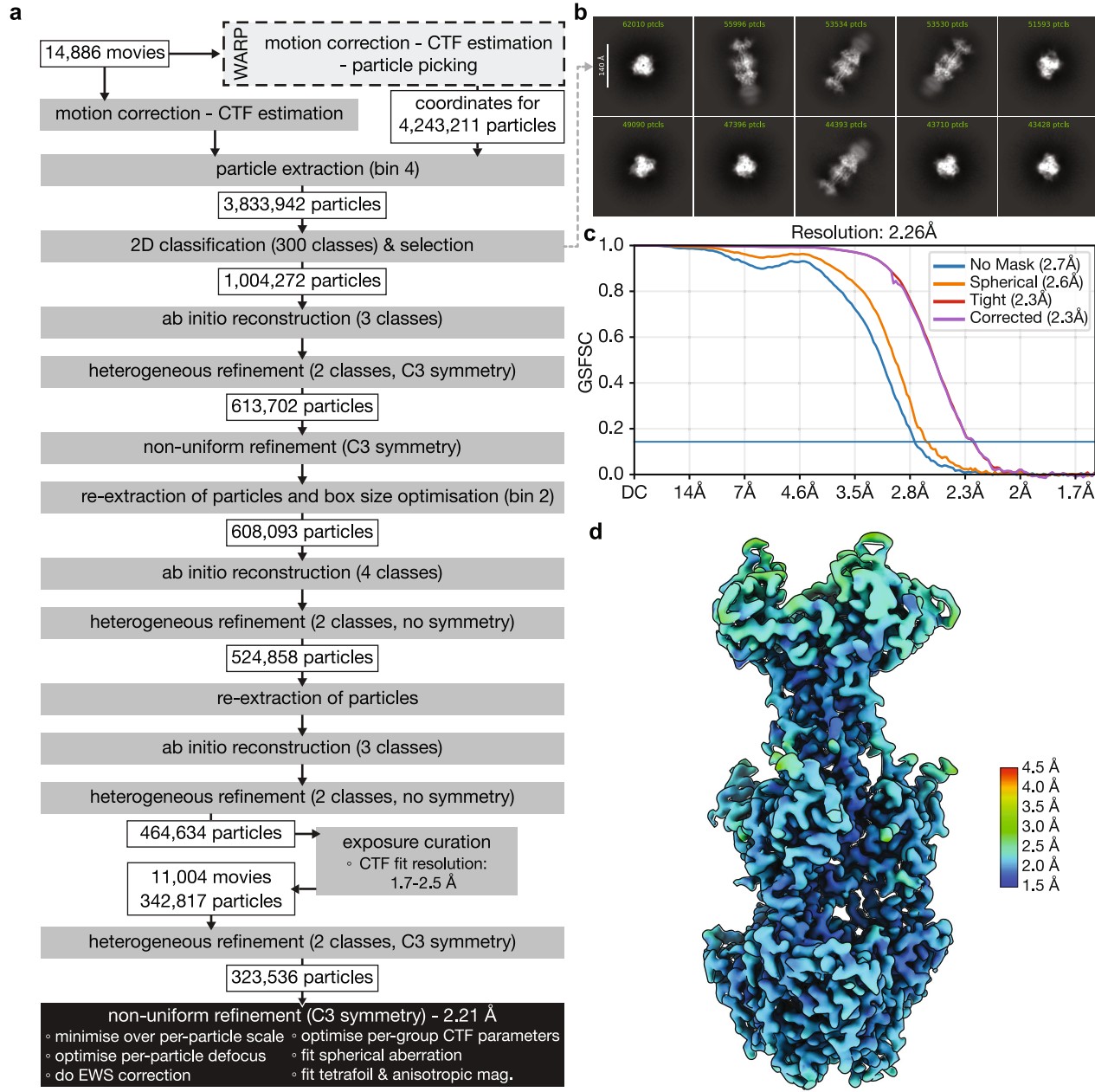

**a**

14,886 movies

WARP: motion correction - CTF estimation - particle picking

motion correction - CTF estimation

coordinates for 4,243,211 particles

particle extraction (bin 4)

3,833,942 particles

2D classification (300 classes) & selection

1,004,272 particles

ab initio reconstruction (3 classes)

heterogeneous refinement (2 classes, C3 symmetry)

613,702 particles

non-uniform refinement (C3 symmetry)

re-extraction of particles and box size optimisation (bin 2)

608,093 particles

ab initio reconstruction (4 classes)

heterogeneous refinement (2 classes, no symmetry)

524,858 particles

re-extraction of particles

ab initio reconstruction (3 classes)

heterogeneous refinement (2 classes, no symmetry)

464,634 particles

exposure curation
◦ CTF fit resolution: 1.7-2.5 Å

11,004 movies
342,817 particles

heterogeneous refinement (2 classes, C3 symmetry)

323,536 particles

non-uniform refinement (C3 symmetry) - 2.21 Å
◦ minimise over per-particle scale  ◦ optimise per-group CTF parameters
◦ optimise per-particle defocus  ◦ fit spherical aberration
◦ do EWS correction  ◦ fit tetrafoil & anisotropic mag.

**b**

62010 ptcls | 55996 ptcls | 53534 ptcls | 53530 ptcls | 51593 ptcls
49090 ptcls | 47396 ptcls | 44393 ptcls | 43710 ptcls | 43428 ptcls

140 Å

**c**

Resolution: 2.26Å

No Mask (2.7Å)
Spherical (2.6Å)
Tight (2.3Å)
Corrected (2.3Å)

GSFSC

DC  14Å  7Å  4.6Å  3.5Å  2.8Å  2.3Å  2Å  1.7Å

**d**

4.5 Å
4.0 Å
3.5 Å
3.0 Å
2.5 Å
2.0 Å
1.5 Å

**Extended Data Fig. 9 | HSV-2 gB in postfusion conformation. (a)** Processing pipeline, outlining individual steps with particle numbers and options used deviating from default settings. Except initial steps using WARP[51], all steps were done using CryoSPARC[52]. **(b)** 2D classes showing the 10 most populated classes. **(c)** Fourier shell correlation (FSC) for the density map corresponding to the postfusion structure is shown. Resolution was calculated using an FSC threshold of 0.143. **(d)** Final map coloured by local resolution (FSC threshold 0.143) using a range of 1.5–4.5 Å.

## Extended Data Table 1 | Interface analysis of Nb1_gbHSV with gB of HSV-1 and HSV-2

**HSV-1 gB : Nb1_gbHSV**

Hydrogen bonds

| | gB protomer A | Atom | Distance (Å) | Nb1_gbHSV | Atom |
|---|---|---|---|---|---|
| 1 | HIS 311 | [NE2] | 3.87 | TYR 39 | [ OH ] |
| 2 | LYS 349 | [NZ ] | 2.85 | ASP 55 | [ OD2] |
| 3 | ARG 318 | [NH1] | 2.95 | SER 58 | [ OG ] |
| 4 | ARG 318 | [NH2] | 2.93 | SER 58 | [ OG ] |
| 5 | ARG 318 | [NH1] | 3.62 | THR 59 | [ O ] |
| 6 | GLY 121 | [O ] | 3.04 | ARG 102 | [ NH1] |
| 7 | GLY 121 | [O ] | 3.35 | ARG 102 | [ NH2] |
| 8 | GLU 286 | [OE2] | 2.82 | ARG 100 | [ NE ] |
| 9 | GLU 286 | [OE2] | 2.77 | ARG 100 | [ NH1] |
| 10 | GLY 292 | [O ] | 2.9 | ARG 102 | [ NH1] |
| 11 | TYR 296 | [OH ] | 3.03 | ARG 102 | [ N ] |
| 12 | SER 313 | [O ] | 2.83 | THR 52 | [ OG1] |
| 13 | ASP 317 | [OD1] | 2.82 | LYS 66 | [ NZ ] |
| 14 | ASP 317 | [OD2] | 2.98 | LYS 66 | [ NZ ] |

Salt bridges

| | gB protomer A | Atom | Distance (Å) | Nb1_gbHSV | Atom |
|---|---|---|---|---|---|
| 1 | LYS 349 | [NZ ] | 3.85 | ASP 55 | [ OD1] |
| 2 | LYS 349 | [NZ ] | 2.85 | ASP 55 | [ OD2] |
| 3 | GLU 286 | [OE2] | 2.82 | ARG 100 | [ NE ] |
| 4 | GLU 286 | [OE2] | 2.77 | ARG 100 | [ NH1] |
| 5 | ASP 317 | [OD1] | 2.82 | LYS 66 | [ NZ ] |
| 6 | ASP 317 | [OD2] | 2.98 | LYS 66 | [ NZ ] |

Hydrogen bonds

| | gB protomer B | Atom | Distance (Å) | Nb1_gbHSV | Atom |
|---|---|---|---|---|---|
| 1 | TYR 655 | [OH ] | 3.13 | ALA 26 | [ O ] |
| 2 | GLN 658 | [N ] | 2.88 | SER 27 | [ O ] |
| 3 | HIS 657 | [ND1] | 3.53 | MET 104 | [ SD ] |
| 4 | SER 656 | [O ] | 2.99 | ARG 29 | [ N ] |
| 5 | ASN 585 | [O ] | 3.32 | ARG 29 | [ NH2] |
| 6 | TYR 655 | [O ] | 3.11 | ARG 29 | [ NH2] |
| 7 | GLU 651 | [OE2] | 2.86 | ARG 56 | [ NH1] |
| 8 | GLU 651 | [OE2] | 2.86 | ARG 56 | [ NH2] |
| 9 | TYR 640 | [OH ] | 3.26 | ARG 102 | [ NH2] |

Salt bridges

| | gB protomer B | Atom | Distance (Å) | Nb1_gbHSV | Atom |
|---|---|---|---|---|---|
| 1 | GLU 651 | [OE2] | 2.86 | ARG 56 | [ NH1] |
| 2 | GLU 651 | [OE2] | 2.86 | ARG 56 | [ NH2] |

**HSV-2 gB : Nb1_gbHSV**

Hydrogen bonds

| | gB protomer A | Atom | Distance (Å) | Nb1_gbHSV | Atom |
|---|---|---|---|---|---|
| 1 | GLY 116 | [ O ] | 3.43 | ARG 102 | [ NH1] |
| 2 | GLY 116 | [ O ] | 3.27 | ARG 102 | [ NH2] |
| 3 | ASP 280 | [ OD2] | 3.55 | ARG 100 | [ NH2] |
| 4 | GLU 281 | [ OE2] | 2.89 | ARG 100 | [ NE ] |
| 5 | GLU 281 | [ OE2] | 2.76 | ARG 100 | [ NH2] |
| 6 | GLY 287 | [ O ] | 2.82 | ARG 102 | [ NH2] |
| 7 | TYR 291 | [ OH ] | 3.13 | ARG 102 | [ N ] |
| 8 | THR 307 | [ O ] | 3.20 | TYR 39 | [ OH ] |
| 9 | SER 308 | [ O ] | 2.77 | THR 52 | [ OG1] |
| 10 | SER 308 | [ OG ] | 3.49 | TYR 39 | [ OH ] |
| 11 | HIS 306 | [ NE2] | 3.48 | TYR 39 | [ OH ] |
| 12 | ARG 313 | [ NH2] | 2.86 | THR 59 | [ O ] |
| 13 | ALA 310 | [ N ] | 3.05 | ASN 60 | [ OD1] |
| 14 | ARG 313 | [ NE ] | 2.91 | ASN 60 | [ OD1] |

Salt bridges

| | gB protomer A | Atom | Distance (Å) | Nb1_gbHSV | Atom |
|---|---|---|---|---|---|
| 1 | ASP 280 | [ OD2] | 3.55 | ARG 100 | [ NH2] |
| 2 | GLU 281 | [ OE2] | 2.89 | ARG 100 | [ NE ] |
| 3 | GLU 281 | [ OE2] | 2.76 | ARG 100 | [ NH2] |

Hydrogen bonds

| | gB protomer B | Atom | Distance (Å) | Nb1_gbHSV | Atom |
|---|---|---|---|---|---|
| 1 | TYR 652 | [ OH ] | 3.89 | GLY 28 | [ N ] |
| 2 | SER 653 | [ O ] | 2.87 | ARG 29 | [ N ] |
| 3 | ASN 582 | [ O ] | 3.24 | ARG 29 | [ NH1] |
| 4 | TYR 652 | [ O ] | 3.04 | ARG 29 | [ NH1] |
| 5 | GLU 648 | [ OE2] | 2.9 | ARG 56 | [ NH1] |
| 6 | GLU 648 | [ OE2] | 2.82 | ARG 56 | [ NH2] |
| 7 | TYR 637 | [ OH ] | 3.09 | ARG 102 | [ NH1] |
| 8 | TYR 652 | [ OH ] | 2.69 | ALA 26 | [ O ] |
| 9 | GLN 655 | [ N ] | 2.9 | SER 27 | [ O ] |
| 10 | HIS 654 | [ NE2] | 3.83 | TRP 101 | [ O ] |

Salt bridges

| | gB protomer B | Atom | Distance (Å) | Nb1_gbHSV | Atom |
|---|---|---|---|---|---|
| 1 | GLU 648 | [ OE2] | 2.9 | ARG 56 | [ NH1] |
| 2 | GLU 648 | [ OE2] | 2.82 | ARG 56 | [ NH2] |

List of hydrogen bonds and salt bridges detailing donor and acceptor atoms and distances per protomer for HSV-1 and -2 gB.

**Extended Data Table 2 | Cryo-EM data collection, refinement and validation statistics**

| | #1 prefusion stabilised gB (PDB ID 9Q9L) | #2 prefusion stabilised gB Nb1_gbHSV bound (PDB ID 9Q9N) | #3 prefusion HSV-2 gB (PDB ID 9Q9S) | #4 postfusion HSV-2 gB (PDB ID 9IH8) |
|---|---|---|---|---|
| **Data collection and processing** | | | | |
| Magnification | 130,000 | 105,000 | 105,000 | 105,000 |
| Voltage (kV) | 300 | 300 | 300 | 300 |
| Electron exposure (e–/Å$^2$) | 47.38 | 45.83 | 43.84 | 43.84 |
| Defocus range (μm) | -0.5 - -2.75 | -0.5 - -2.75 | -0.5 - -2.75 | -0.5 - -2.75 |
| Pixel size (Å) | 0.66 | 0.83 | 0.83 | 0.83 |
| Symmetry imposed | C3 | C3 | C3 | C3 |
| Initial particle images (no.) | 2,925,770 | 956,386 | 3,901,343 | 3,833,942 |
| Final particle images (no.) | 275,470 | 394,394 | 158,509 | 323,536 |
| Map resolution (Å) | 2.64 | 2.95 | 2.7 | 2.21 |
| FSC threshold | 0.143 | 0.143 | 0.143 | 0.143 |
| Map resolution range (Å) min, 25th percentile, median, 75th percentile, max | 1.578, 2.496, 3.030, 4.260, 23.126 | 1.814, 2.630, 3.184, 4.634, 8.119 | 1.795, 2.570, 3.101, 5.051, 27.873 | 1.832, 2.048, 2.364, 3.204, 27.139 |
| **Refinement** | | | | |
| Initial model used (PDB code) | ModelAngelo | ModelAngelo | ModelAngelo | ModelAngelo |
| Software | TEMPy-ReFF | TEMPy-ReFF | TEMPy-ReFF | TEMPy-ReFF |
| Model resolution (Å) | 2.6 | 2.9 | 2.7 | 2.2 |
| FSC threshold | 0.143 | 0.143 | 0.143 | 0.143 |
| Model resolution range (Å) | - | - | - | - |
| Map sharpening $B$ factor (Å$^2$) | -102 | -128.8 | -88.2 | -70.7 |
| Model composition | | | | |
| Non-hydrogen atoms | 16,537 | 19,029 | 15,417 | 14,241 |
| Protein residues | 2,079 | 2,409 | 1,935 | 1,764 |
| Ligands | 0 | 24 (NAG) | 24 (NAG) | 18 (NAG) |
| factors (Å$^2$) | | | | |
| Protein (min/max/mean) | 21.35/315.81/56.60 | 19.74/315.81/59.59 | 9.67/146.03/44.04 | 0.79/197.10/38.46 |
| Ligand | - | 55.71/315.81/192.18 | 62.54/315.81/176.46 | 49.27/315.81/156.70 |
| R.m.s. deviations | | | | |
| Bond lengths (Å) | 0.009 | 0.011 | 0.011 | 0.011 |
| Bond angles (°) | 1.995 | 1.983 | 2.040 | 1.901 |
| Validation | | | | |
| MolProbity score | 0.62 | 0.65 | 0.79 | 0.51 |
| Clashscore | 0.12 | 0.42 | 0.19 | 0.04 |
| Poor rotamers (%) | 0.40 | 0.69 | 1.09 | 0.20 |
| Ramachandran plot | | | | |
| Favored (%) | 97.92 | 97.99 | 96.90 | 98.63 |
| Allowed (%) | 2.08 | 1.89 | 2.94 | 1.37 |
| Disallowed (%) | 0.00 | 0.13 | 0.16 | 0.00 |

| | #5 Apo gB Pre- & postfusion conformation | #6 gB + Nb1_gbHSV Pre- & postfusion conformation | #7 gB + Nb2_gbHSV Pre- & postfusion conformation | #8 gB + Nb3_gbHSV Pre- & postfusion conformation | #9 gB + Nb4_gbHSV Pre- & postfusion conformation |
|---|---|---|---|---|---|
| **Data collection and processing** | | | | | |
| Magnification | 105,000 | 105,000 | 105,000 | 105,000 | 105,000 |
| Voltage (kV) | 300 | 300 | 300 | 300 | 300 |
| Electron exposure (e–/Å$^2$) | 45.3 | 45.3 | 45.3 | 45.3 | 45.3 |
| Defocus range (μm) | -0.5 - -2.25 | -0.5 - -2.25 | -0.5 - -2.25 | -0.5 - -2.25 | -0.5 - -2.25 |
| Pixel size (Å) | 0.83 | 0.83 | 0.83 | 0.83 | 0.83 |
| Symmetry imposed | C3 | C3 | C3 | C3 | C3 |
| Initial particle images (no.) | 1,610,019 | 852,141 | 1,627,438 | 473,269 | 434,737 |
| Postfusion conformation | | | | | |
| Final particle images (no.) | 275,514 | 83,044 | 56,918 | 65,895 | 151,118 |
| Prefusion conformation | | | | | |
| Final particle images (no.) | 442,978 | 237,867 | 105,077 | 197,544 | 183,229 |
| Map resolution (Å) | 6.9 | 6.9 | 6.9 | 6.9 | 6.9 |
| FSC threshold | 0.143 | 0.143 | 0.143 | 0.143 | 0.143 |

# Reporting Summary

## Statistics

For all statistical analyses, confirm that the following items are present in the figure legend, table legend, main text, or Methods section.

| n/a | Confirmed | |
|---|---|---|
| ☐ | ☒ | The exact sample size (*n*) for each experimental group/condition, given as a discrete number and unit of measurement |
| ☐ | ☒ | A statement on whether measurements were taken from distinct samples or whether the same sample was measured repeatedly |
| ☒ | ☐ | The statistical test(s) used AND whether they are one- or two-sided<br>*Only common tests should be described solely by name; describe more complex techniques in the Methods section.* |
| ☒ | ☐ | A description of all covariates tested |
| ☒ | ☐ | A description of any assumptions or corrections, such as tests of normality and adjustment for multiple comparisons |
| ☐ | ☒ | A full description of the statistical parameters including central tendency (e.g. means) or other basic estimates (e.g. regression coefficient) AND variation (e.g. standard deviation) or associated estimates of uncertainty (e.g. confidence intervals) |
| ☒ | ☐ | For null hypothesis testing, the test statistic (e.g. *F*, *t*, *r*) with confidence intervals, effect sizes, degrees of freedom and *P* value noted<br>*Give P values as exact values whenever suitable.* |
| ☒ | ☐ | For Bayesian analysis, information on the choice of priors and Markov chain Monte Carlo settings |
| ☒ | ☐ | For hierarchical and complex designs, identification of the appropriate level for tests and full reporting of outcomes |
| ☒ | ☐ | Estimates of effect sizes (e.g. Cohen's *d*, Pearson's *r*), indicating how they were calculated |

*Our web collection on statistics for biologists contains articles on many of the points above.*

## Software and code

Policy information about availability of computer code

| Data collection | SerialEM 4.0 |
|---|---|
| Data analysis | Excel 16.16.27(201012) (Microsoft), CryoSPARCv4, WARP (http://www.warpem.com/warp/), ISOLDE 1.8, ChimeraX 1.8, Phenix version 1.20.1-4487, MolProbity 4.5.2, CCP-EM Doppio 0.5.0, coot 0.9.8.93, TEMPyReFF, WAVEcontrol software (Creoptix AG), MO.control (Nanotemper), SnapGene v8.1, Adobe Illustrator 2024, Adobe Photoshop 2024 |

For manuscripts utilizing custom algorithms or software that are central to the research but not yet described in published literature, software must be made available to editors and reviewers. We strongly encourage code deposition in a community repository (e.g. GitHub). See the Nature Portfolio guidelines for submitting code & software for further information.

## Data

Policy information about availability of data

All manuscripts must include a data availability statement. This statement should provide the following information, where applicable:
- Accession codes, unique identifiers, or web links for publicly available datasets
- A description of any restrictions on data availability
- For clinical datasets or third party data, please ensure that the statement adheres to our policy

The EM density maps have been deposited in the Electron Microscopy Data Bank under EMD-52963, EMD-52965, EMD-52863 and EMD-52966. The corresponding models have been deposited in the Protein Data Bank under 9Q9L, 9Q9N, 9IH8, 9Q9S.

In preparation of this manuscript the following, atomic structures have been used that are available on the Protein Data Bank: 1IGT, 6CGR, 6Z9M, 5V2S, 2GUM, 7KDP, 8VG6, 3DUZ. For sequence alignments the following sequences were used that are available on the UniProt database: Q4JR05, P08666, A1Z0P7, P03188, F5HB81, F5HB53, P52352, P36320, P36319.

## Research involving human participants, their data, or biological material

Policy information about studies with human participants or human data. See also policy information about sex, gender (identity/presentation), and sexual orientation and race, ethnicity and racism.

| | |
|---|---|
| Reporting on sex and gender | n/a |
| Reporting on race, ethnicity, or other socially relevant groupings | n/a |
| Population characteristics | n/a |
| Recruitment | n/a |
| Ethics oversight | n/a |

Note that full information on the approval of the study protocol must also be provided in the manuscript.

# Field-specific reporting

Please select the one below that is the best fit for your research. If you are not sure, read the appropriate sections before making your selection.

☒ Life sciences        ☐ Behavioural & social sciences        ☐ Ecological, evolutionary & environmental sciences

For a reference copy of the document with all sections, see nature.com/documents/nr-reporting-summary-flat.pdf

# Life sciences study design

All studies must disclose on these points even when the disclosure is negative.

| | |
|---|---|
| Sample size | No statistical methods were used to predetermine sample size. The data size for cryoEM was determined by the availability of the microscope time and the particle density on the grids. Sufficient cryo-EM data were collected to achieve the reported resolution of map, which is sufficient for model building. The sample sizes for functional assays were chosen to ensure reproducibility. |
| Data exclusions | CryoEM data processing involved removing poor-quality or damaged particles to achieve high resolution maps through pre-established standard data classification procedures. No other data was excluded. |
| Replication | Screening of HSV-1 neutralisation by different nanobodies was done with at least two successful, technical replicates. IC50 value for Nb1_gbHSV was determined from three successful biological replicates with three technical replicates each. Expression analysis for disulfide constructs was determined from three successful biological replicates. Expression and protein conformation analysis of different stabilised constructs of gB is shown by Western Blot analysis and by cryoET as examples of one of at least two successful biological replicates. Binding tests of Nb1_gbHSV to different homologues by cell surface staining of gB was analysed from three successful biological replicates. Affinity measurements of the different nanobodies or controls to gB were determined each from three successful biological replicates. |
| Randomization | This is not relevant to our study, because no grouping was needed. Only during cryoEM data analysis randomisation is used, which is automatically applied by the used software. |
| Blinding | Investigators were not blinded to group allocation, because no grouping was needed for this study. Only during cryoEM data analysis 'blinding' was used by omitting the use of preliminary models and instead using ab-inito models generated by the data itself for all structures. |

# Reporting for specific materials, systems and methods

We require information from authors about some types of materials, experimental systems and methods used in many studies. Here, indicate whether each material, system or method listed is relevant to your study. If you are not sure if a list item applies to your research, read the appropriate section before selecting a response.

## Materials & experimental systems

| n/a | Involved in the study |
|-----|----------------------|
| ☐ | ☒ Antibodies |
| ☐ | ☒ Eukaryotic cell lines |
| ☒ | ☐ Palaeontology and archaeology |
| ☐ | ☒ Animals and other organisms |
| ☒ | ☐ Clinical data |
| ☒ | ☐ Dual use research of concern |
| ☒ | ☐ Plants |

## Methods

| n/a | Involved in the study |
|-----|----------------------|
| ☒ | ☐ ChIP-seq |
| ☒ | ☐ Flow cytometry |
| ☒ | ☐ MRI-based neuroimaging |

## Antibodies

| | |
|---|---|
| Antibodies used | rabbit anti-His6 antibody (abcam, ab9108), anti-rabbit-HRP (Sigma-Aldrich Chemie GmbH, A0545), mouse anti-GAPDH antibody (Sigma-Aldrich Chemie GmbH, G8795), anti-mouse-HRP (Sigma-Aldrich Chemie GmbH, AP181P), Strep-Tactin® HRP (iba Lifesciences, 2-1502-001), SS55 (Gary Cohen, Penn State University), Nanobodies described in this study are named Nb1_gbHSV - Nb17_gbHSV |
| Validation | Reference citations are available on the manufacturers websites for all antibodies. <br> rabbit anti-His6 antibody - https://www.abcam.com/en-us/products/primary-antibodies/6x-his-tag-antibody-ab9108?srsltid=AfmBOor8hj_ep9I-47nUJqsdXpKoUht2Kyg1PFrtymVxI0P4kLDBsjlc <br> anti-rabbit-HRP - https://www.sigmaaldrich.com/DE/de/product/sigma/a0545 <br> mouse anti-GAPDH antibody - https://www.sigmaaldrich.com/DE/de/product/sigma/g8795 <br> anti-mouse-HRP - https://www.sigmaaldrich.com/DE/de/product/mm/ap181p <br> SS55 - characterised in doi:10.1128/jvi.03200-13 <br> Nanobodies Nb1_gbHSV - Nb4_gbHSV were further characterised and their ability to bind gB was validated by grating-coupled interferometry and cryoEM. |

## Eukaryotic cell lines

Policy information about cell lines and Sex and Gender in Research

| | |
|---|---|
| Cell line source(s) | BHK-21 (C13), HEK293T & Vero (CCL-81) - all acquired from ATCC |
| Authentication | Authentication was only done via morphology and growth behaviour in the cell type specific media. |
| Mycoplasma contamination | All cell lines were regularly tested for mycoplasma contamination and all results were negative. |
| Commonly misidentified lines (See ICLAC register) | None commonly misidentified cell lines were used. |

## Animals and other research organisms

Policy information about studies involving animals; ARRIVE guidelines recommended for reporting animal research, and Sex and Gender in Research

| | |
|---|---|
| Laboratory animals | One female alpaca was used, kept at the Alpaca Facility of the Max Planck Institute for Multidisciplinary Sciences (Göttingen). |
| Wild animals | The study did not involve wild animals. |
| Reporting on sex | For generation of nanobodies one female alpaca was immunized. |
| Field-collected samples | The study did not involve samples collected from the field. |
| Ethics oversight | Immunizations and blood sampling of the alpaca has been approved by the animal welfare authority LAVES with the reference numbers 33.9-42502-05-13A351, 33.9-42502-05-17A220, and 33.19-42502-04-22-00210. |

Note that full information on the approval of the study protocol must also be provided in the manuscript.

## Plants

Seed stocks

n/a

Novel plant genotypes

n/a

Authentication

n/a

