## [Peer Review File · Nature]

A nanobody specific to prefusion glycoprotein B neutralises HSV-1 and -2

Corresponding Author: Professor Kay Gruenewald

Version 1:

Reviewer comments:

Referee #1

(Remarks to the Author)

In their manuscript, Vollmer and colleagues stabilized and determined cryo-EM structures of HSV1 gB in its prefusion conformation (after its reconstitution in Peptidiscs). The structure includes membrane-embedded regions and several previously unresolved structural features, which provide insights into the initial conformational changes required for membrane fusion.

The authors also generate nanobodies against gB by injecting gB-containing extracellular vesicles into alpaca. nanobodies capable of recognizing gB on the surface of vesicles were selected and tested for their potential to neutralize HSV1. Only 2 of them out of 17 neutralized the virus.

The structure of the complex between Nb1 and HSV-1 gB was determined by cryo-EM and showed that the nanobody was located at the interface between 3 domains that are close in prefusion conformation. This suggests that Nb1 neutralizes HSV1 by preventing gB from undergoing its fusogenic conformational change.

Nb1 nanobody also binds HSV2 gB and the authors solve the structure of HSV2 both in its postfusion conformation and in its prefusion conformation in association with Nb1.

In conclusion, they state that most of the changes during the structural transition are attributed to DV, which is buried within the prefusion conformation structure. Thus, the differences between the pre- and postfusion structures in terms of antigenic epitopes are minimal, making it difficult to specifically target the prefusion structure.

These findings are very similar to those presented in the back-to-back manuscript although different immunization approaches (in different hosts, nanobodies vs antibodies) were used. Technically, the work is clearly well done.

The prefusion structure presented here is more complete than that presented in the other manuscript as it includes the C-terminal part of the ectodomain and the transmembrane domain. It is worth noting that this work also provides the structures of HSV2 gB. Nevertheless, the prefusion structures of HSV1 or HSV2 gB are not so different from the structure of the prefusion structure of human cytomegalovirus glycoprotein B (Liu et al. 2021).

Overall, the manuscript therefore provides significant data that could have an impact on developing vaccine strategies against human herpesviruses. The epitope they identified that is bound by Nb1_gbHSV is certainly an attractive target.

However, I think the study has some limitations that should at least be discussed.

1) A major bias of this study which is that the authors have only characterized a neutralizing nanobody (Nb1). However, it would be very interesting to understand why the other nanobodies that bind gB are not neutralizing. Do they bind accessible epitopes in the same way in both conformations? Do they bind an epitope that is accessible only on the soluble ectodomain but not on gB on the surface of the virus? Can we explain what will distinguish them from neutralizing antibodies in terms of

recognized epitope?

I am perfectly aware that the authors are not going to produce several other structures between gB and non neutralizing antibodies, but this point must be discussed.

Nevertheless, it is probably possible to test the binding between certain neutralizing or non-neutralizing nanobodies and gB at the surface of purified virions.

2) The authors worked with nanobodies that are very small compared to antibodies. It is not impossible that in the context of an antibody, the steric hindrance much greater than that induced by a nanobody makes that some paratopes would allow to neutralize the virus by interfering with the structural transition of gB and/or its neighbors. This could be mentioned as an alternative to the last sentence of their discussion (Judging from the tight space in this region it is possible, that conventional antibodies would be too big to reach this epitope, especially on the viral envelope).

3) The authors propose a pathway for gB conformational change. Can they compare it to what is proposed for other class III glycoproteins like rhabdovirus glycoproteins (PMID: 28188244, PMID: 32814045, PMID: 32004500) and baculovirus gp64 (which is also a class III but not subject to any pressure from the immune system since the virus infects lepidoptera) whose prefusion structure has just been solved (PMID: 3922737)

Referee #2

(Remarks to the Author)

Comments for the authors

Summary

In this manuscript, Vollmer et al report high-resolution cryoEM structures of the gB fusogen from herpes simplex viruses 1 and 2 (HSV-1 and HSV-2) in two conformational states, prefusion and postfusion, and bound to a neutralizing nanobody (Nb). The new findings in the manuscript can be split into two groups: 1) the new structural knowledge about gB conformations and 2) interactions of Nb with gB and proposed neutralization mechanism.

1) The authors determined four new high-resolution cryoEM structures of the gB fusogen from herpes simplex viruses 1 and 2 (HSV-1 and HSV-2): the mutationally stabilized HSV-1 gB in prefusion state, alone and bound to a nanobody; a wild-type HSV-2 gB in prefusion state bound to the same nanobody, and a wild-type HSV-2 gB in postfusion state. The three prefusion structures determined here resemble the three prefusion gB structures that have been reported previously, a low-resolution structure of HSV-1 gB determined by the same group (ref. 16) and high-resolution structures of gB from a related human cytomegalovirus (HCMV) (ref. 4 and 17). Likewise, the postfusion structure of HSV-2 gB resembles the previously reported structures of HSV-1 gB (refs. 2 and 3). But although the global conformational differences between prefusion and postfusion states of gB have been described previously, the new high-resolution structures reveal local differences between prefusion and postfusion gB as well as between HSV-1 and HSV-2 gB. The authors do a very nice job detailing the differences between all known gB structures from HSV and HCMV, highlighting conformational differences in the fusion loops, the N terminus, among others. These observations, along with the complete "set" of HSV structures will be useful to other researchers in the field.

2) The authors used vesicles derived from gB-expressing cells to immunize alpacas and isolated 17 Nbs. Surprisingly, only one of these showed consistent, strong neutralization of HSV-1 and HSV-2 infection measured by reduction in plaque formation. The structures showed that the Nb bound prefusion gB at the interface spanning between two domains and even made an additional contact to a third domain. Although no binding studies have been carried out, when added to a mixture of prefusion and postfusion HSV-2 gB, the Nb was bound only to the prefusion state and not the postfusion. Therefore, the authors concluded that the NB was prefusion-specific and neutralized infection by stabilizing gB in its prefusion state.

Major concerns:

1. The assertion that Nb is prefusion-specific should be backed up by binding assays with prefusion and postfusion forms. Are the rest of the 16 isolated Nbs prefusion-specific?
2. The authors propose that binding of Nb to the prefusion state stabilizes it and prevents it from refolding into the postfusion state. But aside from the structure, no data supporting this conclusion are provided. Moreover, why is postfusion state of gB is observed when gB and Nb are co-expressed? This would suggest that the stabilization is inefficient. Curiously, the prefusion state of HSV-2 gB is very flexible even when bound to the Nb. Could it be because Nb binding does not stabilize it fully?

Additional concerns:

1. There are no line numbers, which makes references to specific sections cumbersome.
2. End of Intro: Summary paragraph in the introduction would be useful in priming the reader.
3. Design of prefusion stabilizing gB variants: Expression or vesicle formation data for the quadruple mutant is missing.
4. Figure 1d: representative images do not make it clear that the gB mutant containing vesicles are decorated only with prefusion gB. Particularly, H516P mutants looks like it has a mixture of extended and globular glycoproteins on its surface. It would be more convincing if the authors made space for a zoomed-in view of pre- vs postfusion gB and indicated differences in height and width of pre- vs. post. Quantification would be even more convincing.
5. Figure 2: Does fusion loop II contact DV in the prefusion state?
6. Figure 3 Legend: shownsel should be shown.
7. End of Neutralizing activity of Nb1_gbHSV: Were any intermediate conformations found in the dataset?
8. Orchestrated changes for membrane fusion: The description of the envisioned order of events is very confusing. Transitions between prefusion structure and intermediate model could be more clearly illustrated.
9. Extended figures: multiple panels spanning several pages are very confusing to the reader.

(Remarks to the Author)

Vollmer et al describe the cryo-EM structure of the elusive pre-fusion structure of the class III membrane fusion protein gB from herpes-simplex viruses 1 and 2 at 2.6Å resolution or so. They also report the cryo-EM structure to 1.9Å resolution of gB in complex with a neutralizing nanobody targeting a quaternary epitope present only in the pre-fusion conformation, thereby identifying a vulnerability site on the viral glycoprotein that can be used in epitope-based vaccine design approaches. The authors follow up on their initial discovery that cells transfected with full-length gB secrete gB coated vesicles, in which they observed a fraction of gB adopting the typical post-fusion conformation, while another fraction was in a different conformation, assumed to be pre-fusion. They used these vesicles to immunize alpaca and select specific nanobodies, some of which were specific for the pre-fusion form. They also identified that a helix-breaking mutation (H516P) in the central coiled-coil helix of its post-fusion form makes recombinant gB more resistant to spontaneously transiting into the post-fusion form on vesicles secreted from gB transfected cells. They used vesicles carrying this mutant to obtain a 9Å resolution cryo-ET and sub-tomogram averaging map of pre-fusion gB. The individual domains of gB could be docked into this map, providing a pseudo-atomic model. Higher resolution studies were however hampered by the fact that this mutant still switched to the post-fusion conformation when solubilized from the membranes to carry out structural studies. The authors therefore went further in this study by adding additional mutations, one in the C-terminal part of the coiled-coil helix (which remains unaltered in pre-fusion and post-fusion) and stabilized its conformation by completing the heptad repeat pattern of non-polar residues stabilizing this portion of the coiled coil with mutation N709V, and by further identifying two residues to change into cysteine (393 and 532) to engineer a disulfide bond connecting two domains that are close by in the pre-fusion form (as indicated by the cryo-ET structure) but that are far apart in the post-fusion form, which allows them to solubilize the further stabilized gB and make single particle cryo-EM studies at higher resolution.

The authors also show that of 17 nanobodies isolated that were specific for the pre-fusion form, only one was potentially neutralizing. They provide its structures in complex with HSV-1 gB. They show that it is cross-reactive with HSV-2 gB, and use it in co-expression experiments with HSV-2 gB, showing that it binds to a the wild-type pre-fusion form, for which they also determine a cryo-EM structure of the complex to 2.7Å resolution. This structure, which shows more flexibility than the stabilized mutants of gB, shows that the main features are maintained, implying that the mutations that were introduced for stabilization of HSV-1 gB did not induce a major distortion of the molecule. This is a very important internal control.

The structures reported in this manuscript show key novel features that provide insight into the trigger of the gB fusogenic conformational change. One of them is the presence of an alpha-helix in the N-terminal region, which is disordered in the post-fusion form, that wedges between domains DI and DII, in a hinge region that must adopt a different relative conformation between these two domains to drive membrane fusion. Another key feature described here for the first time is the conformation of the gB fusion loops in the pre-fusion form, which is very different than that observed in the post-fusion form. The difference relates to the interactions that these loops make with the membrane proximal external region (MPER), which they contact only in the pre-fusion form.

Overall, the results described in this manuscript are highly significant and provide key new elements to understand the conformational changes that drive membrane fusion for entry of herpes viruses, and provides a handle for developing potential treatments, and the stabilization of specific immunogens for vaccine design.

The mechanism of neutralization described for the nanobody described in this manuscript is reminiscent of the one deployed by antibodies neutralizing rabies virus, which also has a class III fusion protein (see DOI: 10.1126/sciadv.abp9151 and 10.1038/s41467-020-14398-7). It would be worth discussing potential similarities in their neutralization mechanisms with the one described here, as there may be common features that apply to all class III fusion proteins.

While reviewing this manuscript, the following pre-print was deposited in bioRxiv:

<https://doi.org/10.1101/2024.10.23.619923>, which describes a computational design using machine-learning- and structure-guided approaches to stabilize the pre-fusion conformation of the EBV gB ectodomain. In this process, the authors identified stabilized conformational intermediates of EBV gB, which they postulate suggest a plausible mechanism for gB-mediated fusion that may extend to all class III viral fusion proteins. It would be important to discuss the fusion mechanism proposed in Figure 6 with the one proposed in the preprint, as it can be quite relevant. Also, the neutralization mechanism of the nanobody can be addressed in the light of those new data.

I list additional minor issues below, which can help the authors improve the manuscript.

- 1- The description of the structure does not mention the cytosolic/intraviral domain of gB. Is it because it is disordered? Maybe this should be explicitly stated.
- 2- The manuscript should provide a table listing all the polar interactions observed in the nanobody/gB complex, with the proton donor and acceptor atoms of both molecules.
- 3- The identification of a vulnerable site on pre-fusion gB, where the nanobody binds, should be better highlighted in the manuscript. Especially that it is a nanobody targeting a quaternary epitope that is exclusively found in the pre-fusion form. Yet the description that the authors make is a bit confusing. For instance, in page 7, when they state: "The interaction features a total of 21 hydrogen bonds and 8 salt bridges distributed over separate interaction surfaces on DIV and DI of the previous protomer". What do they mean "by the "previous" protomer? Do they mean that one of these patches is in DIV of one protomer the other in DI of an adjacent protomer? But then they say: "In addition, R102 of Nb1_gBHSV reaches into the gap between DI and DIV of adjacent protomers, forming hydrogen bonds with DI and DIII of the same protomer". Please rephrase this paragraph to make it clearer, as these are important features of the nanobody.
- 4- In the description of the HSV-2 gB pre-fusion structure, the authors write: "The prefusion structure contains the majority of the ectodomain with the density discontinuing from residue 683 at the start of DV" does this mean that they see no density for DV at all? It should be specified more clearly. There is no linear gB diagram indicating the residues that are ordered in the structure reported, which would be very useful. This could be added to Figure 2, and also in Fig. ED4b, as in the comment

below about these Figures.

5- Figure 2 would be clearer if a linear diagram color-coded by domains (and with aa numbers of domain boundaries) at the top. On the ribbon diagram (fig. 2a), the protomer on the foreground could be highlighted to better see how the domains intertwine. This could be done, for instance, by having the two in the background shown in washed colors, or by just having them in two shades of gray. The aa numbers of the first and last residues of the polypeptide chains in one of the protomers could be indicated (also at places where there are breaks in the chain, if possible).

6- It would be very useful if the authors could complete Extended Data Figure 4b by adding the HSV-1 gB secondary structure elements on top of the amino acid alignment provided, also color-coding them according to domains.

Version 2:

Reviewer comments:

Referee #1

(Remarks to the Author)

In the revised version of their manuscript, the authors performed the experiments I requested.

Particularly, they performed several experiments for a set of our non-neutralising nanobodies to analyse binding affinities to the two conformations of gB and structurally determined their binding sites.

This is a significant amount of work that strengthens the manuscript.

Interestingly they show that all analysed nonneutralising nanobodies bind an alpha helix located at the apex of the prefusion structure.

My only concern is the conclusion that they draw in line 419

"From this it can be deduced that nanobodies must have the following properties in order to neutralise: (i) Binding to pre-fusion gB, with (ii) sufficient affinity, and (iii) weaker binding to post-fusion, resulting in a ΔG that forms an energy barrier that is sufficient to stabilise gB in prefusion conformation."

This may be true. However, the fact that all their non-neutralizing antibodies bind the same region of gB (which is certainly the most accessible when the glycoprotein is in its pre-fusion conformation on the surface of the virus) suggests that this region could also act as a decoy for the immune system. I think this would be worth adding to the discussion.

Minor point:

In the legend of Supplementary Figure S7, a reference is missing. The authors have left a (REF) in parentheses.

Referee #2

(Remarks to the Author)

Summary of the key results

The authors have done a nice job addressing the comments and, in particular, further characterizing their nanobodies. They have now done binding studies with the seventeen nanobodies (Nbs) isolated against gB. They show that the nanobody Nb1 – that has shown strong neutralization of HSV-1 and HSV-2 as measured by reduction in plaque formation – preferentially binds the prefusion form. Surprisingly, they found that several non-neutralizing Nbs also preferentially bind the prefusion form, albeit with a lower affinity. The authors conclude that the neutralization ability of Nb1 is due to its much higher, picomolar affinity, rather than specificity for the prefusion conformation.

Remaining concerns:

1. In describing the Nb1 mechanism, the authors alternately state that the binding of Nb1 to the prefusion state stabilizes it (line 398) or that it does not stabilize it but instead prevents it from refolding into the postfusion state (lines 376-377).

However, what the authors have conclusively shown is that Nb1 (as well as Nb2-4) preferentially binds the prefusion form.

No assays have been done to measure either stabilization of the prefusion form or pre-to-post refolding. Additionally, no correlation has been established between affinity of an Nb for the prefusion form and its ability to stabilize the prefusion form or prevent conformational changes. Therefore, any conclusions regarding potential mechanism of neutralization are too speculative and should be toned down, to avoid misleading the reader.

2. The authors propose that picomolar affinity of Nb1 rather than its prefusion specificity is the reason for its neutralization activity. This is a compelling idea. How can it be reconciled with the nanomolar affinity of neutralizing Fabs? Does Nb1 have a better neutralizing ability than known neutralizing antibodies? If not, then focus on the prefusion form might not be the best strategy for developing strongest neutralizing agents.

3. The abstract should be modified to better summarize the findings of the manuscript. The most important findings are the isolation of prefusion-specific nanobodies, one of which is neutralizing, and the determination of the epitope of the latter. However, the abstract glances over this, focusing instead on the high-resolution structures that add nice details but are not conceptually new. For example, the meaning of the sentence on lines 38-39 "This mode-of-action explains the basis of neutralization..." is unclear because the previous sentence only describes the epitope.

Additional concerns:

1. Changes in the revised manuscript were not clearly marked in the text, which made reviewing them difficult.

2. Lines 76-79. Here, prefusion-specific Nbs that are non-neutralizing should be mentioned and the proposed neutralization mechanism should be briefly summarized.

3. Lines 444-446. Can authors clarify what the knowledge of the Nb1 epitope enables?

Referee #3

(Remarks to the Author)

Vollmer et al have modified their manuscript by considering my suggestions and those of the other reviewers. Overall, the revised version is clearer and includes more data, in particular, showing where some of the other nanobodies bind, and also showing that the lack of neutralization of the majority of the pre-fusion gB-specific nanobodies is because of their weak affinity (which to me, is not a surprising observation, by the way, as the authors put it).

Highlights of the paper are the identification at high resolution of the pre-fusion conformation of HSV-1 and HSV-2 gB, the description of the interaction of the fusion loops with the membrane-proximal transmembrane region, the identification of the role of the ordered N-terminal alpha-helix in maintaining the pre-fusion conformation, and the discovery of a vulnerability site targeted by the neutralizing nanobody that can be used for epitope-focused vaccine designs. The detailed comparison with the high-resolution structure of gB from the human cytomegalovirus, which was recently published, is also an interesting aspect. Overall, the results described are highly significant and will interest a wide range of readers.

I only address remaining issues here regarding the presentation of their data.

1. In Extended Data Figures 2d, 3d 9d and 10d, it would help the reader if the authors could use the same color spectrum for the same overall resolution range (i.e., from 1.5Å to 5Å resolution, even if some maps do not reach this resolution, to directly compare the corresponding panels. As currently displayed, it falsely seems to suggest that the resolution is higher (mostly blue) in 2d vs 3d, for instance.

2. Figure 4a (line 150) is cited before Fig. 2 (line 154).

3. I was confused with the order of the Extended data Figures. There are no titles in the different pages of the file, and the fourth page appears to correspond to ED Fig. 5, whereas the 5th page would be ED Fig. 4, according to the ED Fig legends. Also, there is no explanation of the meaning of the SMOC analysis, I presume that the score goes from 0 (white) to 1 (dark blue), color coded from light-blue to dark-blue?

4. Supplementary Figure 5 is useful, but its legend is incomplete. It is not explained what the meaning of the bar plot on top of each amino acid is, nor the color-code used. Also, the colored diagram at the bottom-right end of the panel is not explained. Could they be given on a sequence alignment of HSV-1 and HSV-2 gB, with the secondary structure for each sequence provided above and below the sequences? Same for the SMOC scores. In this case, the HSV-2 gB sequence could have two sets of secondary structure symbols one for pre-fusion and another for post-fusion. But this would make it easier to see where there are variations along the aa sequence, and to directly compare the post vs pre-fusion conformations in HSV2 gB, instead of having three different panels that are more difficult to compare.

We thank the referees for their constructive input. We have performed several additional experiments and reworked the manuscript. We included several of our nanobodies in various analyses to understand the difference between neutralisers and non-neutralisers. Therefore, to allow the readers to better follow the individual nanobodies, we decided to give their names as successive numbers Nb1_HSV to Nb17_HSV.

Please find below our point-by-point response.

Referees' comments:

Referee #1 (Remarks to the Author):

In their manuscript, Vollmer and colleagues stabilized and determined cryo-EM structures of HSV1 gB in its prefusion conformation (after its reconstitution in Peptidiscs). The structure includes membrane-embedded regions and several previously unresolved structural features, which provide insights into the initial conformational changes required for membrane fusion.

The authors also generate nanobodies against gB by injecting gB-containing extracellular vesicles into alpaca. nanobodies capable of recognizing gB on the surface of vesicles were selected and tested for their potential to neutralize HSV1. Only 2 of them out of 17 neutralized the virus.

The structure of the complex between Nb1 and HSV-1 gB was determined by cryo-EM and showed that the nanobody was located at the interface between 3 domains that are close in prefusion conformation. This suggests that Nb1 neutralizes HSV1 by preventing gB from undergoing its fusogenic conformational change.

Nb1 nanobody also binds HSV2 gB and the authors solve the structure of HSV2 both in its postfusion conformation and in its prefusion conformation in association with Nb1.

In conclusion, they state that most of the changes during the structural transition are attributed to DV, which is buried within the prefusion conformation structure. Thus, the differences between the pre-and postfusion structures in terms of antigenic epitopes are minimal, making it difficult to specifically target the prefusion structure.

These findings are very similar to those presented in the back-to-back manuscript although different immunization approaches (in different hosts, nanobodies vs antibodies) were used. Technically, the work is clearly well done.

The prefusion structure presented here is more complete than that presented in the other manuscript as it includes the C-terminal part of the ectodomain and the transmembrane domain. It is worth noting that this work also provides the structures of HSV2 gB. Nevertheless, the prefusion structures of HSV1 or HSV2 gB are not so different from the structure of the prefusion structure of human cytomegalovirus glycoprotein B (Liu et al. 2021).

Overall, the manuscript therefore provides significant data that could have an impact on developing vaccine strategies against human herpesviruses. The epitope they identified that is bound by Nb1_gbHSV is certainly an attractive target.

However, I think the study has some limitations that should at least be discussed.

We thank the reviewer for their positive appreciation of the results.

1) A major bias of this study which is that the authors have only characterized a neutralizing nanobody (Nb1). However, it would be very interesting to understand why the other

nanobodies that bind gB are not neutralizing. Do they bind accessible epitopes in the same way in both conformations? Do they bind an epitope that is accessible only on the soluble ectodomain but not on gB on the surface of the virus? Can we explain what will distinguish them from neutralizing antibodies in terms of recognized epitope?

I am perfectly aware that the authors are not going to produce several other structures between gB and non neutralizing antibodies, but this point must be discussed.

Nevertheless, it is probably possible to test the binding between certain neutralizing or non-neutralizing nanobodies and gB at the surface of purified virions.

We thank the reviewer for this comment. In order to address this question, we performed several experiments for a set of our non-neutralising nanobodies to analyse binding affinities to the two conformations of gB (including a known monoclonal antibody for comparison) and structurally determined the binding sites. We have added this information in a new paragraph 'Prefusion specificity and neutralisation' detailing our respective results (pages//8, lines 298-328 in the revised text). The results, at first might seem surprising, as we found that most of our nanobodies have a strong preference for prefusion gB, but still do not neutralise. Nevertheless, we found that not prefusion specificity, but the difference in affinity to the two conformations is the key to neutralisation. The results prompted us to formulate three properties a nanobody has to have in order to be neutralising (from line 419).

From this it can be deduced that nanobodies must have the following properties in order to neutralise: (i) Binding to pre-fusion with (ii) sufficient affinity. (iii) Binding to post-fusion must be weaker, resulting in a ΔG that forms the energy barrier to stabilise gB in prefusion conformation. Presumably, there is a ΔG threshold for a neutralising effect.

2) The authors worked with nanobodies that are very small compared to antibodies. It is not impossible that in the context of an antibody, the steric hindrance much greater than that induced by a nanobody makes that some paratopes would allow to neutralize the virus by interfering with the structural transition of gB and/or its neighbors. This could be mentioned as an alternative to the last sentence of their discussion (Judging from the tight space in this region it is possible, that conventional antibodies would be too big to reach this epitope, especially on the viral envelope).

This is indeed a very good point and we changed the end of the discussion, which now reads (from line 447):

Given the narrow space in this region, it is possible that conventional antibodies would be too large to reach this epitope, especially on the viral envelope. Conversely, it is not impossible that some paratopes in context of an antibody could neutralise the virus by interfering with the structural transition of gB and/or interaction with its neighbours due to the steric hindrance, which is much greater than that of a nanobody.

3) The authors propose a pathway for gB conformational change. Can they compare it to what is proposed for other class III glycoproteins like rhabdovirus glycoproteins (PMID: 28188244, PMID: 32814045, PMID: 32004500) and baculovirus gp64 (which is also a class III but not subject to any pressure from the immune system since the virus infects lepidoptera) whose prefusion structure has just been solved (PMID: 39227374)

This is a helpful comment allowing to discuss that recently in a preprint suggested partially alternative model. As also reviewers 2 and 3 suggested additional aspects to this discussion and the editor suggested to move Figure 6 from main text to the supplementary material (now as new Supplementary Fig. S8). We added a Supplementary Discussion and Supplemental figures S7 & S8 that covers the differences between herpesvirus gB, and the other class III glycoproteins, with an emphasis on the situation in Rhabdoviridae and baculovirus gp64.

Referee #2 (Remarks to the Author):

Comments for the authors

Summary

In this manuscript, Vollmer et al report high-resolution cryoEM structures of the gB fusogen from herpes simplex viruses 1 and 2 (HSV-1 and HSV-2) in two conformational states, prefusion and postfusion, and bound to a neutralizing nanobody (Nb). The new findings in the manuscript can be split into two groups: 1) the new structural knowledge about gB conformations and 2) interactions of Nb with gB and proposed neutralization mechanism.

1) The authors determined four new high-resolution cryoEM structures of the gB fusogen from herpes simplex viruses 1 and 2 (HSV-1 and HSV-2): the mutationally stabilized HSV-1 gB in prefusion state, alone and bound to a nanobody; a wild-type HSV-2 gB in prefusion state bound to the same nanobody, and a wild-type HSV-2 gB in postfusion state. The three prefusion structures determined here resemble the three prefusion gB structures that have been reported previously, a low-resolution structure of HSV-1 gB determined by the same group (ref. 16) and high-resolution structures of gB from a related human cytomegalovirus (HCMV) (ref. 4 and 17). Likewise, the postfusion structure of HSV-2 gB resembles the previously reported structures of HSV-1 gB (refs. 2 and 3). But although the global conformational differences between prefusion and postfusion states of gB have been described previously, the new high-resolution structures reveal local differences between prefusion and postfusion gB as well as between HSV-1 and HSV-2 gB. The authors do a very nice job detailing the differences between all known gB structures from HSV and HCMV, highlighting conformational differences in the fusion loops, the N terminus, among others. These observations, along with the complete “set” of HSV structures will be useful to other researchers in the field.

2) The authors used vesicles derived from gB-expressing cells to immunize alpacas and isolated 17 Nbs. Surprisingly, only one of these showed consistent, strong neutralization of HSV-1 and HSV-2 infection measured by reduction in plaque formation. The structures showed that the Nb bound prefusion gB at the interface spanning between two domains and even made an additional contact to a third domain. Although no binding studies have been carried out, when added to a mixture of prefusion and postfusion HSV-2 gB, the Nb was bound only to the prefusion state and not the postfusion. Therefore, the authors concluded that the NB was prefusion-specific and neutralized infection by stabilizing gB in its prefusion state.

We thank the reviewer for highlighting the relevance of our work.

Major concerns:

1. The assertion that Nb is prefusion-specific should be backed up by binding assays with prefusion and postfusion forms.

This is an important point and we thank the reviewer for addressing it. We performed several experiments to confirm our assumption. First, we modelled Nb1_gbHSV, bound to its main interaction site on DI onto the postfusion conformation of gB. This revealed that an interaction is unlikely, as this would lead to clashes with DV and the neighbouring DI. This is shown in a new Supplementary Fig. S2. In addition, we performed affinity measurements using microscale thermophoresis and determined the affinity for prefusion gB to be \$K_D = 14\$ pM while no affinity was detected for the ectodomain of wild type gB that adopts the postfusion conformation. We also confirmed these results using grating-coupled interferometry. We immobilized either prefusion stabilized gB or the postfusion ectodomain. Also, in this experimental setup, Nb1_gbHSV bound to prefusion gB with pM affinity, while no affinity to postfusion gB could be detected. We included these results in the manuscript in the paragraphs titled ‘Nb1_gbHSV binds an interdomain epitope’ (from line 253) and ‘Prefusion specificity and neutralisation’ (from line 302) detailing these experiments with accompanying Supplementary Fig. S3.

Are the rest of the 16 isolated Nbs prefusion-specific?

This is an excellent question. We selected a subset of non-neutralising Nbs that we investigated further. Tested in our established grating-coupled interferometry setup, all three showed nM affinities to prefusion gB, but none for postfusion gB. At first puzzled by this result we went on to purify full length wild type gB that adopts the postfusion conformation during solubilization and reconstitution into a membrane mimetic disc in order to decrease potential differences between the two purified proteins used for the measurements. Also, with the full-length construct, no affinity could be measured, and the controls bound in the same way as to the ectodomain.

To see if the neutralizing activity could be coupled to the epitope, we generated single particle analysis datasets using a mixture of prefusion stabilised gB and wild type postfusion gB together with the different nanobodies. Difference maps, generated from the refined volumes of the nanobody bound samples against the apo volumes revealed that all analysed non-neutralising nanobodies bind helix α X at the apex of the prefusion structure. The nanobodies were used in a 50-fold molar excess to gB. This probably also allowed very low affinity interactions to postfusion gB that we partially saw in our EM difference maps, but were not detected in GCI. These results are now included in the new paragraph 'Prefusion specificity and neutralisation' (from line 302) as well as Extended Data Fig. 7c-g and are covered in the discussion.

2. The authors propose that binding of Nb to the prefusion state stabilizes it and prevents it from refolding into the postfusion state. But aside from the structure, no data supporting this conclusion are provided. Moreover, why is postfusion state of gB observed when gB and Nb are co-expressed? This would suggest that the stabilization is inefficient. Curiously, the prefusion state of HSV-2 gB is very flexible even when bound to the Nb. Could it be because Nb binding does not stabilize it fully?

We thank the reviewer for these questions. Finding postfusion gB in our protein preps could have different reasons, beginning with the co-expression of the nanobody and gB. It could well be that expression levels did not fully match. In addition, our stable cell lines were not selected and therefore a pool of cells most probably exist expressing either only gB or the nanobody. Finding postfusion gB does not necessarily indicate insufficient stabilization, however going back through the data we found a range of particles that could potentially show gB trapped in intermediate conformations. Unfortunately, we were not able to achieve any useful resolutions to even see the presence or absence of the nanobody (see answer to your question 7). To tone down our statement we have rephrased our hypothesis in the description of the fusion process – former Fig. 6 and now Supplementary Figure 8 and Supplementary Discussion:

Nb1_gbHSV links DI to DIV of neighbouring protomers within the trimer (Fig. 4) and prevents gB to reach the postfusion conformation (Fig. 5) resulting in effective neutralisation (Fig. 1).

Regarding the flexibility of prefusion gB, as detailed in 'Neutralising activity of Nb1_gbHSV' (from line 333) there is still flexibility in the prefusion wild type gB protein, even when bound to the nanobody. We emphasized this point more by adding the following sentence (from line 375):

Therefore, binding of the nanobody does not fully stabilise the structure, but rather prevents the major conformational changes necessary for fusion.

Additional concerns:

1. There are no line numbers, which makes references to specific sections cumbersome.

We have added the line numbers.

2. End of Intro: Summary paragraph in the introduction would be useful in priming the reader. We thank the reviewer for the advice and added the following sentences at the end of the introduction (from line 76):

Here, we report a neutralizing nanobody against gB that is prefusion specific with cross-species activity. Further, we determined the full-length cryoEM structure of gB in its pre- and post-fusion conformation and describe the nanobody binding site and mode of neutralisation.

3. Design of prefusion stabilizing gB variants: Expression or vesicle formation data for the quadruple mutant is missing.

We added the missing information to Extended data figure 1, in addition to a representative cryoET slice of the formed vesicles in comparison to the wild type construct, carrying the same affinity tag on the C-terminus.

4. Figure 1d: representative images do not make it clear that the gB mutant containing vesicles are decorated only with prefusion gB. Particularly, H516P mutants looks like it has a mixture of extended and globular glycoproteins on its surface. It would be more convincing if the authors made space for a zoomed-in view of pre- vs postfusion gB and indicated differences in height and width of pre- vs. post. Quantification would be even more convincing.

We thank the reviewer for making this point. We added zoomed-in views of both conformations, but this alone – in our opinion – was not clear enough. Therefore, we also added pictograms, detailing the shape and height of each conformation and tried to emphasize the height difference in the images with dashed lines. In addition, we performed more experiments in order to confirm our assessment of pre- to postfusion conformation ratios we made, based on our tomography data. For this we collected transmission image data sets of vesicles with gB for each construct and assigned each vesicle to one of three categories – carrying gB only in prefusion, only in postfusion or a mix of both conformations. The data is now included as Extended Data Fig. 1e. This quantification is consistent with our previous assessment, that the individual mutations have a stabilizing effect on the membrane embedded gB.

5. Figure 2: Does fusion loop II contact DV in the prefusion state?

Yes, there is a hydrophobic interaction of F262 of fusion loop II with a pocket created by residues I185, L252 and residues L686 & Y689 (DV) of the next protomer. This is mentioned at the end of the structure description in line 216, although in the previous submission, we forgot to include Y689, which was added now.

6. Figure 3 Legend: shownsel should be shown.

We thank the reviewer pointing out this typo. Somehow the rest of the sentence was deleted and replaced by 'sel'. It now reads (from line 250):

The MPR is not shown to clearly see the hydrophobic footprint of fusion loop I

7. End of Neutralizing activity of Nb1_gbHSV: Were any intermediate conformations found in the dataset?

For the dataset containing Nb1_gbHSV with the stabilized version of HSV-1 gB there were no other conformations found. Nevertheless, this question prompted us to go back to the HSV-2 gB + Nb1_gbHSV dataset. Indeed, going through particles that were sorted out early during 2D classification and heterogeneous refinement we found several, fairly well defined 2D classes that seem to represent shorter versions of the postfusion conformation of gB. Although promising, unfortunately further processing did not result in any interpretable densities, as the structures seem to be too flexible to resolve:

Shown are five refined structures created from subsets of the found particles. The last one has a clear resemblance to postfusion gB and one could speculate that the other ones are intermediate states from pre- to postfusion conformation. Unfortunately, we were not able to improve the resolution of these structures. It is possible that what we see is a continuum of structures trapped in motion. Although this could be an exciting insight we decided this to be too speculative to read more into this.

8. Orchestrated changes for membrane fusion: The description of the envisioned order of events is very confusing. Transitions between prefusion structure and intermediate model could be more clearly illustrated.

We understand that the description was apparently not clear enough. Reviewers 1 and 3 also suggested changes to this section. This and the request of the editor to move Fig. 6 to the supplement prompted us to write a longer supplementary discussion, detailing the individual steps while comparing the mechanism to what is known for other class III viral fusion proteins. This is further supported by an additional Supplementary Figure S7.

9. Extended figures: multiple panels spanning several pages are very confusing to the reader. We are grateful for this insight and shortened longer figures to each fit one page. Along with this and the given size limitations we removed the Viewing Direction Distribution plots from Extended Data Figs. 2, 3, 9 and 10.

Referee #3 (Remarks to the Author):

Vollmer et al describe the cryo-EM structure of the elusive pre-fusion structure of the class III membrane fusion protein gB from herpes-simplex viruses 1 and 2 at 2.6Å resolution or so. They also report the cryo-EM structure to 1.9Å resolution of gB in complex with a neutralizing nanobody targeting a quaternary epitope present only in the pre-fusion conformation, thereby identifying a vulnerability site on the viral glycoprotein that can be used in epitope-based vaccine design approaches.

The authors follow up on their initial discovery that cells transfected with full-length gB secrete gB coated vesicles, in which they observed a fraction of gB adopting the typical post-fusion conformation, while another fraction was in a different conformation, assumed to be pre-fusion. They used these vesicles to immunize alpaca and select specific nanobodies, some of which were specific for the pre-fusion form. They also identified that a helix-breaking mutation (H516P) in the central coiled-coil helix of its post-fusion form makes recombinant gB more resistant to spontaneously transiting into the post-fusion form on vesicles secreted from gB transfected cells. They used vesicles carrying this mutant to obtain a 9Å resolution cryo-ET and sub-tomogram averaging map of pre-fusion gB. The individual domains of gB could be docked into this map, providing a pseudo-atomic model. Higher resolution studies were however hampered by the fact that this mutant still switched to the post-fusion conformation when solubilized from the membranes to carry out structural studies. The authors therefore went further in this study by adding additional mutations, one in the C-terminal part of the coiled-coil helix (which remains unaltered in pre-fusion and post-fusion) and stabilized its

conformation by completing the heptad repeat pattern of non-polar residues stabilizing this portion of the coiled coil with mutation N709V, and by further identifying two residues to change into cysteine (393 and 532) to engineer a disulfide bond connecting two domains that are close by in the pre-fusion form (as indicated by the cryo-ET structure) but that are far apart in the post-fusion form, which allows them to solubilize the further stabilized gB and make single particle cryo-EM studies at higher resolution.

The authors also show that of 17 nanobodies isolated that were specific for the pre-fusion form, only one was potentially neutralizing. They provide its structures in complex with HSV-1 gB. They show that it is cross-reactive with HSV-2 gB, and use it in co-expression experiments with HSV-2 gB, showing that it binds to a the wild-type pre-fusion form, for which they also determine a cryo-EM structure of the complex to 2.7Å resolution. This structure, which shows more flexibility than the stabilized mutants of gB, shows that the main features are maintained, implying that the mutations that were introduced for stabilization of HSV-1 gB did not induce and major distortion of the molecule. This is a very important internal control.

The structures reported in this manuscript show key novel features that provide insight into the trigger of the gB fusogenic conformational change. One of them is the presence of an alpha-helix in the N-terminal region, which is disordered in the post-fusion form, that wedges between domains DI and DII, in a hinge region that must adopt a different relative conformation between these two domains to drive membrane fusion. Another key feature described here for the first time is the conformation of the gB fusion loops in the pre-fusion form, which is very different than that observed in the post-fusion form. The difference relates to the interactions that these loops make with the membrane proximal external region (MPER), which they contact only in the pre-fusion form. Overall, the results described in this manuscript are highly significant and provide key new elements to understand the conformational changes that drive membrane fusion for entry of herpes viruses, and provides a handle for developing potential treatments, and the stabilization of specific immunogens for vaccine design.

The mechanism of neutralization described for the nanobody described in this manuscript is reminiscent of the one deployed by antibodies neutralizing rabies virus, which also has a class III fusion protein (see DOI: 10.1126/sciadv.abp9151 and 10.1038/s41467-020-14398-7). It would be worth discussing potential similarities in their neutralization mechanisms with the one described here, as there may be common features that apply to all class III fusion proteins.

We thank the reviewer for alerting us to this similarity. We compared the neutralizing mechanisms in the discussion of the paper, adding the following (from line 439):

In other class III fusion proteins, structural differences have been shown to allow neutralisation by antibodies specific to prefusion conformations in Rabies virus G protein. Two modalities of neutralisation were described. (i) By targeting an epitope on a single domain that is only fully accessible in prefusion is enough to lock the protein in prefusion conformation, hence inhibiting the fusion process⁴⁴. (ii) Recognition of an epitope that is distributed over three domains, one being on an adjacent protomer prevents the conformational changes required in fusion⁴⁵. Similarly, the epitope we found that is bound by Nb1_gbHSV opens an attractive avenue of targeting the prefusion conformation by linking domains, close in pre- but apart in postfusion.

While reviewing this manuscript, the following pre-print was deposited in bioRxiv: <https://doi.org/10.1101/2024.10.23.619923>, which describes a computational design using machine-learning- and structure-guided approaches to stabilize the prefusion conformation of the EBV gB ectodomain. In this process, the authors identified stabilized conformational intermediates of EBV gB, which they postulate suggest a plausible mechanism for gB-mediated fusion that may extend to all class III viral fusion proteins. It would be important to discuss the fusion mechanism proposed in Figure 6 with the one proposed in the preprint, as it can be quite relevant. Also, the neutralization mechanism of the nanobody can be addressed in the light of those new data.

Due to space limitations and suggested by the editor we moved the former Fig. 6 and description of the fusion mechanism to the now created Supplementary Discussion. This

allowed us to extend the discussion to include a comparison to other class III viral fusion proteins with an additional Supplementary Figure S7 detailing the similarity to GP64. We also referenced and discussed the recently published preprint to put it in context with our and previous findings in this discussion.

Please also see the respective answers to reviewer 1 and 2.

I list additional minor issues below, which can help the authors improve the manuscript.

1- The description of the structure does not mention the cytosolic/intraviral domain of gB. Is it because it is disordered? Maybe this should be explicitly stated.

This is a good point and we added the following sentence at the end of the description of the membrane embedded parts (from line 238):

The cytosolic/intraviral domain was not resolved in our structure, which could be due to the missing membrane surface where the C-terminal amphipathic helix h3 would be embedded³.

2- The manuscript should provide a table listing all the polar interactions observed in the nanobody/gB complex, with the proton donor and acceptor atoms of both molecules.

We performed interface analyses using ChimeraX and PDBePISA and created the requested table, which is now Extended Data Table 1.

3- The identification of a vulnerable site on pre-fusion gB, where the nanobody binds, should be better highlighted in the manuscript. Especially that it is a nanobody targeting a quaternary epitope that is exclusively found in the pre-fusion form. Yet the description that the authors make is a bit confusing. For instance, in page 7, when they state: “The interaction features a total of 21 hydrogen bonds and 8 salt bridges distributed over separate interaction surfaces on DIV and DI of the previous protomer”. What do they mean “by the “previous” protomer? Do they mean that one of these patches is in DIV of one protomer the other in DI of an adjacent protomer? But then they say: “In addition, R102 of Nb1_gbHSV reaches into the gap between DI and DIV of adjacent protomers, forming hydrogen bonds with DI and DIII of the same protomer”. Please rephrase this paragraph to make it clearer, as these are important features of the nanobody.

We changed the text to make it more understandable (from line 258):

The interaction features a total of 23 hydrogen bonds and 8 salt bridges distributed over separate interaction surfaces on DIV and DI of the adjacent protomer, resulting in 1260 Å² of buried surface area on gB (Fig. 4c-e, Extended Data Table 1). In addition, R102 of Nb1_gbHSV reaches into the gap between DIV and DI, forming hydrogen bonds with DI and DIII of the same protomer.

In addition, we changed Figure 4b-e to specify the protomer of the domains involved in the interactions and the legend accordingly.

4- In the description of the HSV-2 gB pre-fusion structure, the authors write: “The prefusion structure contains the majority of the ectodomain with the density discontinuing from residue 683 at the start of DV” does this mean that they see no density for DV at all? It should be specified more clearly.

Indeed, the density ‘breaks off’ about 15 residues after the start of DV, potentially due to the flexibility of this region. We changed the text to emphasize this more (from line 364):

The prefusion structure contains the majority of the ectodomain with the density discontinuing from residue 683, leaving all but the first 15 residues of DV unresolved.

Interestingly, comparing the HSV-1 and -2 gB structures in this region, HSV-2 features a short helical segment (aa667-671) that is not seen in HSV-1.

There is no linear gB diagram indicating the residues that are ordered in the structure reported, which would be very useful. This could be added to Figure 2, and also in Fig. ED4b, as in the comment below about these Figures.

We added the linear diagrams in Figs. 2 and 5. In an additional Extended Data Figure 4 we show in linear diagrams the resolved secondary structural elements combined with the fit scores of the model to EM density and domain boundaries for all solved structures.

5- Figure 2 would be clearer if a linear diagram color-coded by domains (and with aa numbers of domain boundaries) at the top. On the ribbon diagram (fig. 2a), the protomer on the foreground could be highlighted to better see how the domains intertwine. This could be done, for instance, by having the two in the background shown in washed colors, or by just having them in two shades of gray. The aa numbers of the first and last residues of the polypeptide chains in one of the protomers could be indicated (also at places where there are breaks in the chain, if possible).

We thank the reviewer for the suggestion. We added the linear diagrams to Figs. 2 and 4 as requested. Please see also answer to question 4.

6- It would be very useful if the authors could complete Extended Data Figure 4b by adding the HSV-1 gB secondary structure elements on top of the amino acid alignment provided, also color-coding them according to domains.

We added an additional Extended Data Fig. 4 for this. Please see answer to question 4.

Referee 1

In the revised version of their manuscript, the authors performed the experiments I requested.

Particularly, they performed several experiments for a set of our non-neutralising nanobodies to analyse binding affinities to the two conformations of gB and structurally determined their binding sites.

This is a significant amount of work that strengthens the manuscript.

Interestingly they show that all analysed nonneutralising nanobodies bind an alpha helix located at the apex of the prefusion structure.

My only concern is the conclusion that they draw in line 419

"From this it can be deduced that nanobodies must have the following properties in order to neutralise: (i) Binding to pre-fusion gB, with (ii) sufficient affinity, and (iii) weaker binding to post-fusion, resulting in a ΔG that forms an energy barrier that is sufficient to stabilise gB in prefusion conformation."

This may be true. However, the fact that all their non-neutralizing antibodies bind the same region of gB (which is certainly the most accessible when the glycoprotein is in its pre-fusion conformation on the surface of the virus) suggests that this region could also act as a decoy for the immune system. I think this would be worth adding to the discussion.

This is indeed a good point and we added the following to the discussion starting from line 539 (all line numbers refer to the version with track changes):

Furthermore, the analysis of non-neutralising nanobodies revealed that all target the α X helix at the apex of gB (Extended Data Fig. 6e-g), which is readily accessible in prefusion conformation on the surface of the virus and could therefore act as a decoy for the immune system while vulnerable epitopes are shielded by glycosylation (Extended Data Fig. 7). Binding affinity measurements of these nanobodies (Supplementary Fig. S4) also showed that even a strong binding preference for the prefusion form does not necessarily correlate with efficient neutralisation activity (Extended Data Fig. 1).

Minor point:

In the legend of Supplementary Figure S7, a reference is missing. The authors have left a (REF) in parentheses.

We thank the reviewer for spotting this mistake and added the missing reference.

Referee 2

Summary of the key results

The authors have done a nice job addressing the comments and, in particular, further characterizing their nanobodies. They have now done binding studies with the seventeen nanobodies (Nbs) isolated against gB. They show that the nanobody Nb1 – that has shown strong neutralization of HSV-1 and HSV-2 as measured by reduction in plaque formation – preferentially binds the prefusion form. Surprisingly, they found that several non-neutralizing Nbs also preferentially bind the prefusion form, albeit with a lower affinity. The authors conclude that the neutralization ability of Nb1 is due to its much higher, picomolar affinity, rather than specificity for the prefusion conformation.

Remaining concerns:

1. In describing the Nb1 mechanism, the authors alternately state that the binding of Nb1 to the prefusion state stabilizes it (line 398) or that it does not stabilize it but instead prevents it from refolding into the postfusion state (lines 376-377).

It is true that we did not use the term 'stabilise' consistently here in respect to mutational stabilisation only, but also for prevention of pre- to postfusion refolding. We changed the text to make this clearer from line XYZ:

lines 479-483

'Therefore, binding of the nanobody does not fully stabilise the structure, but rather prevents the major conformational changes necessary for fusion.'

Now reads:

*Therefore, binding of the nanobody does not completely fix gB in a single position like the stabilising mutations, as areas such as the transmembrane region remain flexible, but rather seems to prevent the major conformational changes required for fusion. Some of the prefusion specific features, resolved in our **mutationally** stabilised structure are also found in wild type gB (Supplementary Fig. S2b).*

lines 545-548

'From this it can be deduced that nanobodies must have the following properties in order to neutralise: (i) Binding to pre-fusion gB, with (ii) sufficient affinity, and (iii) weaker binding to post-fusion, resulting in a ΔG that forms an energy barrier that is sufficient to **stabilise** gB in prefusion conformation.'

Now reads:

*From this it can be deduced that nanobodies must have the following properties in order to neutralise: (i) Binding to pre-fusion gB, with (ii) sufficient affinity, and (iii) weaker binding to post-fusion, resulting in a ΔG that forms an energy barrier that is sufficient to **hold** gB in prefusion conformation.*

However, what the authors have conclusively shown is that Nb1 (as well as Nb2-4) preferentially binds the prefusion form. No assays have been done to measure either stabilization of the prefusion form or pre-to-post refolding. Additionally, no correlation has been established between affinity of an Nb for the prefusion form and its ability to stabilize the prefusion form or prevent conformational changes. Therefore, any conclusions regarding potential mechanism of neutralization are too speculative and should be toned down, to avoid misleading the reader.

We thank the reviewer for the comment and suggestion. It is true that we did not directly measure the pre-to-post refolding in absence/presence of the nanobody to prove the suggested mode of action. We therefore toned down our proposed mechanism of neutralisation by introducing the following changes in the text:

lines 445-449

'The bound epitope spread over different domains that are close in prefusion conformation, suggests that the neutralising activity is achieved by preventing gB to undergo the necessary conformational changes for the fusion process. To **prove** this hypothesis and to show the cross-

species activity we co-expressed wild type HSV-2 gB with the nanobody in stable transduced HEK293T cells³³.’

Now reads:

*The bound epitope spread over different domains that are close in prefusion conformation, suggests that the neutralising activity is achieved by preventing gB to undergo the necessary conformational changes for the fusion process. To **support** this hypothesis and to show the cross-species activity we co-expressed wild type HSV-2 gB with the nanobody in stable transduced HEK293T cells³³.*

lines 521-523

‘The fact that the nanobody is able to fix gB in its prefusion conformation (Fig. 5) explains the inhibition of HSV-1 infection (Fig. 1) by preventing gB from carrying out its membrane fusion activity.’

Now reads:

The fact that prefusion gB is only found with the nanobody bound (Fig. 5) implies the prevention of the major conformational changes, necessary during HSV-1 infection, and therefore would explain the neutralising activity of Nb1_gbHSV (Fig. 1).

2. The authors propose that picomolar affinity of Nb1 rather than its prefusion specificity is the reason for its neutralization activity. This is a compelling idea. How can it be reconciled with the nanomolar affinity of neutralizing Fabs? Does Nb1 have a better neutralizing ability than known neutralizing antibodies? If not, then focus on the prefusion form might not be the best strategy for developing strongest neutralizing agents.

The reviewer has an important point here. It is most probably a combination of prefusion specificity and affinity plus the targeted epitope that separates Nb1_gbHSV in terms of neutralising activity to the non-neutralising nanobodies. To emphasize this more we added the following sentence from line 418:

Taken together the observed difference between neutralising and non-neutralising nanobodies seems to be a combination of high affinity and specificity to the prefusion form of gB, and the targeted epitope.

Another point to be considered is the smaller size of the nanobody compared to full antibodies which could facilitate accessibility of the bound epitope which is mentioned in the discussion which we extended to support this point.

lines 579-581

Given the narrow space in this region, it is possible that conventional antibodies may be too large to access this epitope, especially on the viral envelope.

Now reads:

*Given the narrow space in this region, it is possible that conventional antibodies may be too large to access this epitope, especially on the viral envelope **as shown for the Fab of antibody SS55 which has a higher neutralising activity than the full IgG²¹.***

3. The abstract should be modified to better summarize the findings of the manuscript. The most important findings are the isolation of prefusion-specific nanobodies, one of which is neutralizing, and the determination of the epitope of the latter. However, the abstract glances over this, focusing instead on the high-resolution structures that add nice details but are not conceptually new. For example, the meaning of the sentence on lines 38-39

"This mode-of-action explains the basis of neutralization..." is unclear because the previous sentence only describes the epitope.

We agree with the reviewer that findings about the prefusion specific nanobody are not well presented in the abstract. We changed the abstract which now reads:

From line 25-41:

The nine human herpesviruses, including Herpes Simplex Virus 1 and 2 (HSV-1/-2), Human Cytomegalovirus (HCMV), and Epstein-Barr Virus (EBV) present a significant burden to global public health¹. Their envelopes contain at least 10 different glycoproteins, necessary for host cell tropism, attachment and entry². The best conserved glycoprotein, glycoprotein B (gB), is essential as it performs membrane fusion by undergoing extensive rearrangements from a prefusion to postfusion conformation. Currently, there are no antiviral drugs targeting gB or neutralising antibodies directed against its prefusion form, due to the difficulty to structurally determine and utilise this metastable conformation. Here we show the isolation of prefusion-specific nanobodies, one of which exhibits strong neutralising and cross-species activity. By mutational stabilisation we solved the HSV-1 gB full-length prefusion structure which allowed determining the bound epitope. Our analyses revealed the membrane-embedded regions of gB and previously unresolved structural features^{3,4}, including a novel fusion loop arrangement, providing insights into initial conformational changes required for membrane fusion. Binding an epitope spanning three domains only proximal in the prefusion state, the nanobody kept wild type HSV-2 gB in this conformation and enabled determining its native prefusion structure. This also suggests the mode-of-neutralisation and an attractive avenue for antiviral interventions.

Additional concerns:

1. Changes in the revised manuscript were not clearly marked in the text, which made reviewing them difficult.

We apologise for this mistake.

2. Lines 76-79. Here, prefusion-specific Nbs that are non-neutralizing should be mentioned and the proposed neutralization mechanism should be briefly summarized.

We agree with the reviewer and have rearranged and added to the Introduction the following from line 89:

Here, we report a neutralising nanobody against gB that is prefusion specific with cross-species activity. Further, we determined the full-length high-resolution cryoEM structure of gB in its pre- and post-fusion conformation and describe the nanobody epitope and mode of neutralisation.

Now reads:

Further, we determined the full-length high-resolution cryoEM structure of gB in its pre- and post-fusion conformation and describe the nanobody epitope and mode of neutralisation. The topology of the conformational epitope suggests that binding of the nanobody prevents the conformational change to the postfusion form, necessary for membrane fusion, which in turn would explain the neutralising activity. Moreover, we demonstrate several prefusion specific, but non-neutralising nanobodies that target a different epitope at the apex of prefusion gB, indicating that prefusion specificity does not imply neutralising activity.

3. Lines 444-446. Can authors clarify what the knowledge of the Nb1 epitope enables?

This refers to the following sentence in the manuscript: '*Similarly, the epitope we found that is bound by Nb1_gbHSV opens an attractive avenue of targeting the prefusion conformation by*

linking domains, close in pre- but apart in postfusion.' Although the surfaces of the individual domains on gB do not markedly change during fusion as described in lines X-Z, the arrangement of the domains in relation to each other does significantly. Therefore, targeting conformational epitopes that cross domain borders opens a way to target and lock the prefusion form of gB. A similar mode of action was seen for an antibody targeting the prefusion form of Rabies glycoprotein G. Therefore, we cited this paper to support our conclusion. In order to make this clearer we changed the sentence to the following:

From line 575:

Similarly, the epitope bound by Nb1_gbHSV opens an attractive avenue for neutralisation by binding conformational epitopes that span multiple domains that are only close in pre- but apart in postfusion.

Referee 3

Vollmer et al have modified their manuscript by considering my suggestions and those of the other reviewers. Overall, the revised version is clearer and includes more data, in particular, showing where some of the other nanobodies bind, and also showing that the lack of neutralization of the majority of the pre-fusion gB-specific nanobodies is because of their weak affinity (which to me, is not a surprising observation, by the way, as the authors put it).

Highlights of the paper are the identification at high resolution of the pre-fusion conformation of HSV-1 and HSV-2 gB, the description of the interaction of the fusion loops with the membrane-proximal transmembrane region, the identification of the role of the ordered N-terminal alpha-helix in maintaining the pre-fusion conformation, and the discovery of a vulnerability site targeted by the neutralizing nanobody that can be used for epitope-focused vaccine designs. The detailed comparison with the high-resolution structure of gB from the human cytomegalovirus, which was recently published, is also an interesting aspect. Overall, the results described are highly significant and will interest a wide range of readers.

I only address remaining issues here regarding the presentation of their data.

1. In Extended Data Figures 2d, 3d 9d and 10d, it would help the reader if the authors could use the same color spectrum for the same overall resolution range (i.e., from 1.5Å to 5Å resolution, even if some maps do not reach this resolution, to directly compare the corresponding panels. As currently displayed, it falsely seems to suggest that the resolution is higher (mostly blue) in 2d vs 3d, for instance.

This is a good point and we changed the corresponding panels. The shown resolution range is now 1.5 – 4.5 Å. We also updated the FSC curves and the reached resolutions, that were also used for the PDB submission. Please note that the FSC calculations produced before by the final refinement jobs resulted in marginally higher resolutions compared to the values produced by the final FSC calculations. We changed the numbers in the manuscript accordingly.

2. Figure 4a (line 150) is cited before Fig. 2 (line 154).

We agree that this is confusing. The figure citation in (previously line 150 now line 230) refers to Extended Data Fig. 4a and not main text Fig. 4a. Because Extended Data Fig. 4 was

moved to the Supplementary Information we exchanged the figure reference to 'Extended Data Fig. 3d & Supplementary Fig. S2'.

3. I was confused with the order of the Extended data Figures. There are no titles in the different pages of the file, and the fourth page appears to correspond to ED Fig. 5, whereas the 5th page would be ED Fig. 4, according to the ED Fig legends. Also, there is no explanation of the meaning of the SMOC analysis, I presume that the score goes from 0 (white) to 1 (dark blue), color coded from light-blue to dark-blue?

We thank the reviewer for spotting this mistake. The ED Figs. 4 and 5 were in the wrong order. Due to size restrictions we decided to move the SMOC analysis together with the structural alignment to the supplementary information (now Supplementary Fig. S2), where we also extended the figure legend to give a better explanation of the SMOC analysis. This now reads:

Manders' Overlap Coefficient (SMOC) analysis and structural alignment

a SMOC analysis showing a sequence-based local estimate of the fit quality of the atomic model to the corresponding map (0 – 1, meaning no fit to perfect fit) for every amino acid of each chain of each protein⁶⁵. To emphasize the differences in the calculated range, only values ranging from 0.6 to 0.9 are depicted as shades of blue. Unmodeled parts of the structure which have no SMOC score, are shown in grey.

b Structure alignment analysis showing secondary structure elements of gB from HSV-1 in post-³ and prefusion (Nb1_gbHSV bound) conformation and of HSV-2 in post- and prefusion form. Domain boundaries are shown by colours as in Fig. 2. N-glycosylation sites are shown with black background, while stabilising mutations used in HSV-1 prefusion gB are shown in bold red letters. Regions comprising the fusion loops as well as α -helix X are marked. Given the symmetric nature of the pre and post fusion trimer only the secondary structure for chain A is plotted.

4. Supplementary Figure 5 is useful, but its legend is incomplete. It is not explained what the meaning of the bar plot on top of each amino acid is, nor the color-code used. Also, the colored diagram at the bottom-right end of the panel is not explained. Could they be given on a sequence alignment of HSV-1 and HSV-2 gB, with the secondary structure for each sequence provided above and below the sequences? Same for the SMOC scores. In this case, the HSV-2 gB sequence could have two sets of secondary structure symbols one for pre-fusion and another for post-fusion. But this would make it easier to see where there are variations along the aa sequence, and to directly compare the post vs prefusion conformations in HSV2 gB, instead of having three different panels that are more difficult to compare.

We extended the legend of this Figure (now Supplementary Fig. 5) to explain the diagram at the end of the figure and the conservation score better and we added a colour key and two references concerning the conservation score that is shown as bar plots for each residue.

This now reads:

Clustal Omega multiple sequence alignment. Conservation score^{66,67} for each position is shown as coloured bars followed by the consensus sequence. VZV: varicella zoster virus; HSV-2: herpes simplex virus 2; HSV-1: herpes simplex virus 1; EBV: Epstein-Barr virus; KSHV: Kaposi's sarcoma-associated herpesvirus; HCMV: human. Cytomegalovirus; HHV7: human herpesvirus 7; HHV6A, B: human herpesvirus 6A, B. Residues interacting with one

Nb1_gbHSV are marked with red boxes (gB protomer a) and green boxes (gB protomer b). UniProt IDs of the respective sequences are shown at the end. To illustrate the inter-protomer interaction a top view of gB is shown on the bottom right with protomer a in green and b in red and Nb1_gbHSV in antique white.